# CULTURAL AND LINGUISTIC DIVERSITY IMPROVES VISUAL REPRESENTATIONS

## ABSTRACT

Computer vision often treats perception as objective, and this assumption gets reflected in the way that datasets are collected and models are trained. For instance, image descriptions in different languages are typically assumed to be translations of the same semantic content. However, work in cross-cultural psychology and linguistics has shown that individuals differ in their visual perception depending on their cultural background and the language they speak. In this paper, we demonstrate significant differences in semantic content across languages in both dataset and model-produced captions. When data is multilingual as opposed to monolingual, captions have higher semantic coverage on average, as measured by scene graph, embedding, and linguistic complexity. For example, multilingual captions have on average 21.8% more objects, 24.5% more relations, and 27.1% more attributes than a set of monolingual captions. Moreover, models trained on content from different languages perform best against test data from those languages, while models trained on multilingual content perform consistently well across all evaluation data compositions. Our research provides implications for how diverse modes of perception can improve image understanding.

## 1 INTRODUCTION

There's an old adage that goes, *"a picture is worth a thousand words."* These *"thousands words"* represent the myriad ways different people can interpret an image. Differing interpretations result from a diverse set of informational cues: emotional cues (Gallup et al., 2014), social cues (Watson & Platt, 2012; Gallup et al., 2012), visual elements (e.g., optical blur; (Sprague et al., 2016)) and, of course, factual details. People also incorporate prior experiences, communicative intent (Joo et al., 2014; Park et al., 2019), and contextual information (Yorzinski & Platt, 2014). Moreover, previous work suggests that the way someone perceives and describes the world is also influenced by their culture (Oyserman & Lee, 2008; Masuda et al., 2008; Nisbett & Masuda, 2003) as well as the language they speak (Langacker, 1993; Boroditsky et al., 2003). All of this indicates that different people can look at the same image and yet focus on completely different aspects.

Computer vision models are currently not trained in a way that reflects or aligns with this human condition (Stretcu et al., 2023). Today, the predominant paradigm for training vision models relies on large-scale image-description pairs mined from the internet (Radford et al., 2021; Li et al., 2022). These pairs are scraped and annotated without attention to the underlying humans' visual preferences or their inherent biases (Röttger et al., 2021). Even models finetuned for applications like image captioning rely on human annotated datasets curated without explicitly considering how annotators perceive scenes (Berg et al., 2012; Hutchinson et al., 2022).

The implicit assumption made in this paradigm is that *perception is objective*. This suggests that regardless of who describes an image, the semantic concepts covered in the description should highlight roughly the same salient parts of the image. However, this stands against decades of research in cross-cultural psychology (Hall, 1976). Human studies find evidence that perception varies among individuals, especially between cultures (Nisbett & Masuda, 2003). These factors alter the way individuals attend to different parts of an image and consequently which components they mention when talking about what they see. Studies of different cultures usually end up having language as a factor, because people from different cultures often speak different languages. In fact, many cultural artifacts are emergent in language; for example, Japanese has several more levels of formality (e.g.,

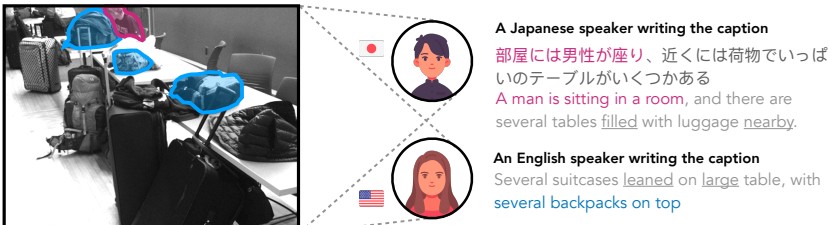

Figure 1: A Japanese speaker and an English speaker in our human evaluation caption the images differently based on several factors such as linguistic constraints and background context. Red and blue highlights in text and image indicate content that is uniquely captured by each speaker.

honorifics (Harada, 1976)), Mandarin infuses Confucian values such as "tranquility" not present in other languages (Wen et al., 2023), and German's complex morphosyntactic system provides events with nuanced understanding of spatial relationships (Prange & Schneider, 2021).

In this paper, drawing from cultural and linguistic factors, we investigate variations in visual perception by examining differences in the semantic coverage of image captions between multiple languages. We study these differences across all the ways that image captions are produced in computer vision: human-annotated, model-produced, and commercial API-produced. We define and evaluate 4 metrics for semantic coverage over 2 datasets, 3 models, and 1 commercial API across 7 languages.

We show—against the current prevailing assumption in computer vision—that people who speak different languages mention different semantic content when describing images, and that these differences are often reflected by models. In general, multilingual captions cover significantly more semantic concepts than monolingual ones. When represented as a joint scene graph (Schuster et al., 2015), a set of multilingual captions have on average 21.8% more objects, 24.5% more relations, and 27.1% more attributes than a set of monolingual captions (Table 2). Comparing on linguistic measures, multilingual captions expand the range of object abstraction/concreteness by 12.2%, analytic expression by 45.0%, and color by 37.5% (Table 5). Providing different captions as training datasets for models leads the model to learn and reproduce levels of abstraction present in the data—image captioning models finetuned on monolingual datasets perform best when evaluated on test sets with descriptions collected in that monolingual language. Moreover, those trained on multilingual data perform consistently well across *all* test sets. As support for the cultural basis for observed differences, our results corroborate findings in cross-cultural psychology (Nisbett & Masuda, 2003; Koo et al., 2018). In a human image captioning study, Japanese individuals tended to mention background objects more frequently than American individuals, who focused more on the foreground objects. In general, across multiple experiments, we observe higher concept overlap between Western languages than between Western and Eastern languages and surprisingly, also between Eastern languages (Tables 3, 7). These results demonstrate the important role of human cultural and linguistic subjectivity in computer vision and contributes a unique computer vision perspective to the long line of research on the subjectivity of perception.

## 2 RELATED WORK

Psychologists, anthropologists, and philosophers provide strong evidence that perception is subjective. Studies have ranged broadly from organizational principles of perception and cognition: from Gestalt psychology (Wertheimer, 1912) and Universal Grammar (Chomsky, 1986), to cognitive realities of one's perceptual experience (Wittgenstein, 1953). Even the perception of length (Segall et al., 1967), geometrical intuition (Pedersen & Wheeler, 1983; Dawson et al., 1973), and depth (Jahoda & McGurk, 1974) vary across people.

Cross-cultural psychology observes that many features of perception vary between individuals from different cultures (Ceněk & Sasinka, 2020). To highlight these differences, anthropologists (Hall, 1976) have introduced the concept of "analytic" versus "holistic" societies. This concept has been broadly used to compare "Westerners" with "Easterners". The Western self is analytical: it is com-

posed of fixed attributes and can move between settings without an alteration. Meanwhile, the Eastern self is holistic: the self is connected to others and their surrounding context (Hall, 1976). Philosophers posit that the Eastern self understands themselves "in terms of their relation to the whole, such as the family, society, Tao Principle, or Pure Consciousness" (Munro, 1985).

Evidence suggests that the differences in cultural selves can alter the way individuals attend to and understand visual content and how they talk about what they see (Nisbett & Masuda, 2003). For instance, when looking an image, Westerners tend to concentrate on a focal object and focus on analyzing its attributes and categorization. Attributions tend to focus exclusively on the object (Masuda & Nisbett, 2001); for example, "the tree is blooming tall" or "that person looks angry". Meanwhile, East Asians are more likely to attend to a broader perceptual and conceptual field, noticing relationships and changes and grouping objects based on resemblance rather than category membership (Koo et al., 2018). For example, an American person might describe an image as "Two old people on a grey park bench", focusing on the objects ("people", "bench") and their attributes ("old", "grey"). While viewing the same image, a Japanese person might see "Two friends laughing over a joke", focusing instead on the relationships ("friends") and actions ("laughing").

Since culture is a difficult component to quantify and isolate, our work uses language as a proxy to study differences between people describing the same image (Goldstein, 1957; Laesser et al., 2014). In addition, language also introduces expressive constraints that incentivize speakers to structure information in a certain way as to avoid sounding unnatural.

In computer vision, recent models have demonstrated the ability to perform an array of incredible tasks (Ramesh et al., 2022; Ouyang et al., 2022). Most of these capabilities are learned by vision models from large swathes of vision-language data scraped from the internet (Radford et al., 2021; Srinivasan et al., 2021). The shift from class labels to image-text pairs (Thomee et al., 2016; Schuhmann et al., 2022; Erdem et al., 2022) opens up computer vision to greater expression of perceptual subjectivity. As such, disregarding the cultural background of language produced to describe images can unduly overrepresent a set of perceptual behaviors over others. Troublingly, vision-language datasets remain largely unrepresentative of most non-Western populations, or are too small or poor-quality for the large scale training of today's complex models. Datasets without perceptual diversity risk contributing towards the homogenization of dominant perceptual modes (Bommasani et al., 2021) into an "algorithmic monoculture" (Kleinberg & Raghavan, 2021). This harms progress towards important human-centric (Berg et al., 2012; Wilchek et al., 2023), decolonization (Bird, 2020), diversity, and equity goals (Hershcovich et al., 2022) for AI models.

Computer vision has begun its first steps towards understanding human perception in a machine learning context, such as in object saliency (Spain & Perona, 2008; Elazary & Itti, 2008; Berg et al., 2012; Yun et al., 2013; Tavakoli et al., 2017), user-centric vision modeling (Stretcu et al., 2023), and task ambiguity (Hutchinson et al., 2022). There have also been attempts to build geographically diverse image datasets (Atwood et al., 2020; Schumann et al., 2021). Some have provided culturally diverse knowledge ontologies for image datasets (Liu et al., 2021). However, our work posits that it is also important to diversely represent the perceptual styles underlying image descriptions.

## 3 ANALYSIS OF DATASETS, MODELS, AND APPLICATIONS

Our paper investigates the following key research question: *Are there differences in the semantic concepts described by vision-language datasets, models, and applications across languages?* The answer to this question could inform how we sample pre-training data for large-scale vision-language models, as well as who we recruit to annotate images with descriptions when finetuning for applications like image captioning. This question challenges the prevailing assumption in computer vision that the contents of an image are objective. Popular datasets such as CIFAR, ImageNet, COCO, PASCAL-VOC have traditionally treated supervised labels as objective (Gordon et al., 2021). If this is so, then monolingual semantic content across languages should be roughly similar and multilingual datasets should not be significantly different from their particular monolingual counterparts. Our analysis suggests that this "objectivity assumption" may not hold.

In order to test the "objectivity assumption", we analyze three stages of today's computer vision modeling pipeline: from data to model to application. We analyze the descriptions generated by human annotators in datasets, annotations generated by models, and by available commercial APIs. We measure the semantic coverage of concepts mentioned in these descriptions to quantify differ-

ences across languages, in particular between monolingual descriptions versus multilingual ones. We limit our experiments to 7 languages.

## 3.1 SOURCES OF IMAGE-LANGUAGE DATA

**Datasets.** We examine the Crossmodal ('XM') dataset (Thapliyal et al., 2022), which contains image descriptions in 36 languages over 3.6k geographically diverse images. Moreover, to provide some support that the observed trends may also occur in humans, we conduct an image captioning human evaluation to collect descriptions for 30 Visual Genome (Krishna et al., 2016) images from 10 English speakers in the United States and 10 Japanese speakers in Japan, recruited from Prolific.

**Models.** We analyze how models trained with multilingual data describe images across different languages. Specifically, we study LLaVA (Liu et al., 2023) to produce multilingual captions for images. Although LLaVA is trained with English data, it inherits probable limited multilingual capabilities from its large language model (LLM) component, LLaMA (Touvron et al., 2023). In total, we generate 64.8k captions for 3.6k Crossmodal images in 6 languages with 3 captions each. LLaVA allows us to study language factors across descriptions because the vision representations are constant across text prompts in different languages (see Appendix D.1 for details on probing multilingual behavior). To provide an English-only baseline, we also compare against descriptions generated by BLIP2 (Li et al., 2023a) and GIT (Wang et al., 2022).

**APIs.** We use the Google Vertex API to generate multilingual captions for images – 75.6k captions for 3.6k Crossmodal images in 7 languages, and 42k captions for 2k Visual Genome images. These outputs reflect established standards and have gone through extensive quality checks. This is our main focus because of its stability and control across languages.

**Languages.** We focus on English, French, German, Russian, Chinese, Japanese, and Korean.[1] Together, these 7 languages encompass a wide range of typologically diverse linguistic families. Moreover, the speakers of these languages originate from a large variety of cultures and experiences. To conduct our experiment, we require the consistent application of linguistic tools such as parsers, which are more mature for English than other languages. Hence, for our analysis, we translate all descriptions, human or model produced, into English with GPT-4 (OpenAI, 2023).

To verify that the translations preserve salient conceptual information, we conduct a human evaluation where we recruit 15 users to rate 30 translations across each of the 6 non-English languages. Participants mark the translations with an average score of $4.68 \pm 0.60$ on a 1-to-5 Likert scale, where 1 is "entirely inaccurate", 5 is 'entirely accurate / perfect translation", and 4 is "most of the information is preserved (in the translation)". In other words, participants marked $98.42\%$ of important information in a visual scene (objects, spatial relations, color, etc.) as faithfully preserved across translation (see Appendix C for more details on the translation prompt and evaluation).

## 3.2 MEASURING SEMANTIC COVERAGE

We measure semantic coverage through four mechanisms: **(A)** complexity of a scene graph, **(B)** representational diversity, **(C)** linguistic diversity, and **(D)** ground truth coverage.

**(A) Complexity of a scene graph.** A scene graph $\mathcal{G}$ is defined as a list of tuples of the form $(\text{object}, \text{attribute})$ or $(\text{subject}, \text{predicate}, \text{object})$. Scene graphs are a common representation to encapsulate the first-order relations between objects in a scene (Johnson et al., 2015b; Choi et al., 2022; Ji et al., 2020).

We use scene graphs to measure the semantic coverage of each individual description. First, we parse the descriptions in our data sources into scene graphs using FLAN-T5 (Chung et al., 2022) model fine-tuned on the FACTUAL-MR dataset.[2] We found that LLM-based parsers were better able to resolve complex semantic relationships which often arose in the descriptions than conventional syntax-based parsers (Schuster et al., 2015). Next, we use the following metrics to measure semantic coverage in the scene: object count (number of objects mentioned), relationship count (number of pairwise objects connected by a relationship), and attribute count (number of attributes mentioned).

---

[1] We do not include LLaVA's Korean captions because they are nearly always incoherent and inaccurate.

[2] The finetuned model is already available online (Li et al., 2023b).

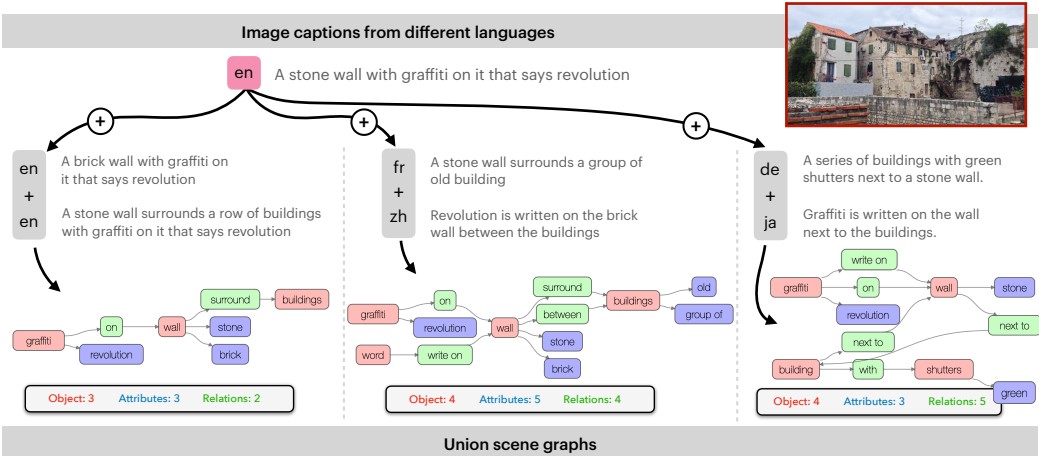

Figure 2: Captions of an image in different languages and their scene graphs, when unioned together produce more varied and complex scene graphs for multilingual distributions than monolingual ones.

We can also leverage scene graphs to understand the overlap between two descriptions. The union of two scene graphs $\mathcal{G}_1 \cup \mathcal{G}_2$ represents the concepts shared between them: $\mathcal{G}_1$ and $\mathcal{G}_2$. Often, descriptions use different words (e.g. "person" and "man") when referring to the same object. To avoid double counting, we canonicalize the concepts using WordNet path similarity (Miller, 1995) and cosine similarity between concept embeddings (Reimers & Gurevych, 2019) to merge concepts which have different text but ground to the same concept. Multiple relationships between two entities are similarly merged into a single edge using the same process.

We define the information gained from a scene graph $\mathcal{G}$ to $\mathcal{G}_+$ across metric $M$ as $M(\mathcal{G} \cup \mathcal{G}_+) - M(\mathcal{G})$. The metric of an intersection can be computed using inclusion-exclusion as follows: $M(\mathcal{G}_1) + M(\mathcal{G}_2) - M(\mathcal{G}_1 \cup \mathcal{G}_2) = M(\mathcal{G}_1 \cap \mathcal{G}_2)$.

**B** **Representational diversity.** The second measure of semantic coverage we use is representational diversity in the embedding space occupied by a description. Let $E$ be a text embedding model (Reimers & Gurevych, 2019) and $\mathbb{P}$ be a set of natural language text. Let $\mathbb{Q} = \{E(\boldsymbol{p}) : \boldsymbol{p} \in \mathbb{P}\}$. We measure its coverage as the maximum pairwise cosine distance, $coverage(\mathbb{Q}) = \max\left(\left\{1 - \frac{\boldsymbol{a} \cdot \boldsymbol{b}}{\|\boldsymbol{a}\|\|\boldsymbol{b}\|} : \{\boldsymbol{a}, \boldsymbol{b}\} \in \mathbb{Q} \times \mathbb{Q}\right\}\right)$. Consider $\mathbb{P}_A \sim D_A$ and $\mathbb{P}_{A \cup B} \sim D_A \cup D_B$, text samples drawn from the monolingual language A text distribution $D_A$ and a multilingual distribution $D_A \cup D_B$ comprising jointly of language A and B, respectively. If $\mathbb{E}\left[coverage(\mathbb{Q}_{A \cup B})\right] > \mathbb{E}\left[coverage(\mathbb{Q}_A)\right]$, then the multilingual distribution $D_A \cup D_B$ has a wider semantic coverage than the monolingual distribution $D_A$, as measured by $E$.

**C** **Linguistic diversity.** Linguistic measures allow us to examine specific dimensions of text content according to linguistic principles. For some metric $M$ which maps text to a real number and some set of text $\mathbb{T}$, we define the semantic coverage of the set as measured by $M$ to be $coverage_M(\mathbb{T}) = \max(\{M(i) : i \in \mathbb{T}\}) - \min(\{M(i) : i \in \mathbb{T}\})$. If $\mathbb{E}[coverage_M(\mathbb{A})] > \mathbb{E}[coverage_M(\mathbb{B})]$ for $\mathbb{A}$ sampled from a multilingual distribution and $\mathbb{B}$ sampled from a monolingual distribution, then we claim that the multilingual distribution is more diverse/rich than the monolingual one as measured by $M$. We use several metrics $M$. *Concreteness* is computed across noun objects and indicates how much a word refers to a perceptible entity (Brysbaert et al., 2014). For instance, "purple" has a high concreteness rating (4.04 on a 5-point scale), whereas "beautiful" has a low one (2.16). *Analytic*, *Clout*, *Authentic*, and *Tone* are psychological measures of logical and hierarchal thinking patterns, social status and confidence, spontaneity, and emotional tone in text, from the Linguistic Inquiry and Word Counts (LIWC) package (Tausczik & Pennebaker, 2010). *Color* is computed as the maximum pairwise distance between colors (using the $\Delta$E algorithm (Sharma et al., 2005), which ensures perceptual uniformity) mentioned across text.

**D** **Ground truth coverage**. Our final metric for semantic coverage uses ground truth data as reference. The Visual Genome dataset ((Krishna et al., 2016)) provides dense annotations: 50 descrip-

tions per image with their associated scene graphs. Given a ground truth scene graph, we measure how many objects are represented in the Visual Genome annotations. Formally, given the ground truth object set $\mathbb{G}$ and the objects mentioned in a caption $\mathbb{C}$, semantic coverage is $\frac{|\mathbb{C} \cap \mathbb{G}|}{|\mathbb{G}|}$.

## 3.3 RESULTS

Across all semantic coverage mechanisms outlined above, we find that multilingual data improves semantic coverage compared to monolingual data. The increases are more pronounced for model generated descriptions than human annotated captions but are visible in both.

**(A) Complexity of a scene graph.**

We run LLaVA and the Google Vertex API inference on the Crossmodal dataset (3.6k images). For the model and the API, sampling an equal number of descriptions from multilingual data yields higher semantic coverage than sampling from a monolingual distribution (Table 2). In fact, despite an expected diminishing-returns trajectory, continuously unioning even a well-developed existing scene graph with a new language's scene graph expands it (Figure 3). Since image captions are typically short and targeted to focus on the more salient parts of an image, this continual increase in the unique objects suggests that the salient objects are not self-evident across languages, as assumed by prior work (Berg et al., 2012),

Table 1: Scene graph metrics across human evaluation captions shows that unioning from 5 English and 5 Japanese scene graphs is richer than ones from 10 English or 10 Japanese.

|  |  | en | ja | en,ja |
|---|---|---|---|---|
| **Metric** | **Objects** | 9.13 | 8.97 | 9.90 |
|  | **Relations** | 9.63 | 9.30 | 10.37 |
|  | **Attributes** | 9.17 | 8.93 | 9.77 |

but rather vary widely. We see the same pattern in scene graphs constructed from our human evaluation (Table 1) in which sampling captions from across Japanese and English annotations increases scene graph size over sampling individually from Japanese or English annotations. Moreover, a manual inspection of the captions (Figure 8) suggests that the captions roughly echo the predictions from cross-cultural perceptual psychology – Japanese captions tend to mention background objects and information more than English ones. The identification of such patterns across multiple sources (humans and models) provides some support against the prevailing "objectivity" assumption.

We also compare these values against an English baseline consisting of the union between a Vertex/LLaVA, GIT, and BLIP2 parsed scene graph. The baseline is 'ambitious' in the sense that the Vertex/LLaVA, GIT, and BLIP2 do not share the same architecture or training data. The only thing they have in common is that their training data uses English data. Therefore, looking at Figure 3, it is noteworthy that adding additional Vertex captions in different languages continues to increase the coverage of objects, attributes, and relationships. This increase is approximately the same as if we had started with a baseline using just Vertex/LLaVA's English descriptions.

The concepts mentioned by GIT and BLIP2 have a surprisingly low content intersection with all of the monolingual Vertex scene graphs (Table 3). This suggests that although additional models increase the number of concepts by about as much as additional languages, the content of these

Table 2: Scene graph metrics across Vertex and LLaVA captions in different languages show that multilingual scene graph unions are richer than monolingual ones. "Monolingual" represents scene graph unions from 3 captions within the same language. "Multilingual" represents scene graph unions from 3 captions in the given languages. "mm" refers to the multimodel baseline. Increases are relative to the English average. **(A)**

|  |  | Monolingual | | | | | | | Multilingual | | | |
|---|---|---|---|---|---|---|---|---|---|---|---|---|
|  |  | en | de | fr | ru | zh | ja | ko | en,fr,zh | fr,de,ru | all | mm |
| **Vertex** | **Objects** | 3.65 | 3.51 | 3.60 | 3.86 | 3.46 | 3.13 | 3.18 | 4.31 | 4.25 | 5.93 | 4.63 |
|  | **Relations** | 2.96 | 2.83 | 2.89 | 3.20 | 2.68 | 2.37 | 2.47 | 3.60 | 3.56 | 6.08 | 3.64 |
|  | **Attributes** | 1.67 | 1.67 | 1.79 | 1.86 | 1.66 | 1.59 | 1.62 | 2.13 | 2.15 | 3.34 | 2.19 |
| **LLaVA** | **Objects** | 4.54 | 5.05 | 5.26 | 4.52 | 4.54 | 3.32 |  | 5.87 | 6.02 | 8.44 | 6.65 |
|  | **Relations** | 3.79 | 4.21 | 4.42 | 3.67 | 3.66 | 2.40 |  | 4.84 | 4.97 | 7.92 | 3.42 |
|  | **Attributes** | 2.75 | 3.47 | 3.50 | 2.76 | 3.25 | 2.70 |  | 4.10 | 4.07 | 6.98 | 2.88 |

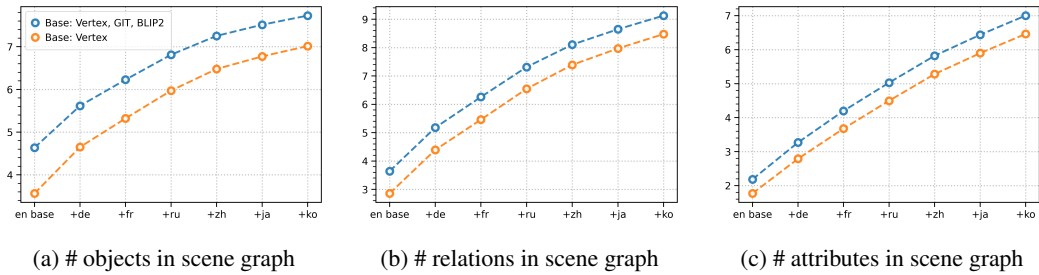

| (a) # objects in scene graph | (b) # relations in scene graph | (c) # attributes in scene graph |

Figure 3: Scene graphs of captions unioned cumulatively from different languages lead to more coverage in objects, relations, and attributes. For more graphs, see Appendix D.3. **A**

Table 3: Intersection sizes between 3 unioned monolingual Vertex captions and an English multimodel baseline (a unioned BLIP2 ∪ GIT scene graph, held constant across all languages) are both relatively **small** and **smaller for Asian than European languages**. All relationships between European languages in red and Asian languages cells in blue are statistically significant with Bonferroni correction. The 'mm' column includes the size of the unioned GIT and BLIP model for reference. **A**

| | | Language | | | | | | | |
|---|---|---|---|---|---|---|---|---|---|
| | | **en** | **de** | **fr** | **ru** | **zh** | **ja** | **ko** | **mm** |
| **Metric** | Objects | 1.96 | 1.92 | 1.93 | 1.97 | 1.85 | 1.73 | 1.76 | 3.59 |
| | Relations | 0.79 | 0.76 | 0.74 | 0.78 | 0.70 | 0.62 | 0.64 | 2.51 |
| | Attributes | 0.44 | 0.36 | 0.37 | 0.42 | 0.37 | 0.37 | 0.33 | 1.45 |

contributed concepts are very different. We even find low content intersection between any two languages, measured using the scene graphs extracted from Vertex monolingual scene graphs (Table 12), demonstrating that content distributions across languages are meaningfully separate. Interestingly, we observe "Western" languages (English, German, French, Russian) tend to cover more similar concepts than to "Eastern" languages (Chinese, Japanese, Korean) (Tables 3 and 12).

**B** **Representational diversity.** When measuring semantic coverage in embedding space, we find that groups of multilingual captions have larger embedding space coverage than monolingual ones. The mean coverage point cloud of three embedded English Vertex captions, for instance, almost doubles from 0.19 to 0.37 when replacing two of them with embedded French and Chinese Vertex captions (Table 4).

We observe similar patterns in the human-labeled Crossmodal dataset, although expectedly with weaker signals (Table 13). For European languages, using a multilingual set over a monolingual one noticeably increases the mean coverage, from 0.38 for monolingual English to 0.46 for English and Chinese. This effect is less prominent for Asian languages.

**C** **Linguistic diversity.** We find, across all linguistic metrics but to varying degrees, that multilingual descriptions increase the linguistic diversity coverage over monolingual data for Vertex (Table 5), and generally for LLaVA (Table 14) and Crossmodal (Table 15). This suggests that multilingual data vary more in the form of their descriptions, and also interestingly perceive/mention a wider range of color.

**D** **Ground truth coverage.** Sampling the same number of multilingual descriptions compared to monolingual data increases the mean semantic coverage of Visual Genome-annotated objects (Table 6), a pattern echoing the results in **A**. Interestingly, when English, French, and Chinese descriptions are merged, their coverage increases when compared to using only English descriptions from 3.70 to 4.31 objects per image. However, their intersection with Visual Genome ground truth objects only increases from 3.49 to 3.84. This suggests that the Visual Genome dataset itself is 'perceptually limited', despite being a canonical densely annotated dataset. This unaccounted $4.31 - 3.84 = 0.47$ objects likely refer to objects which have not been documented in Visual Genome. We provide several examples to support this hypothesis in Appendix D.7 and discuss the implications in

Table 4: Embeddings from multilingual captions have a larger mean coverage than embeddings from monolingual captions. Crossmodal provides at minimum two captions per language per image, so we adjust the multilingual compositions from 3 to 2 accordingly. **B**

| | | | Monolingual | | | | | | Multilingual | | |
|---|---|---|---|---|---|---|---|---|---|---|---|
| | en | de | fr | ru | zh | ja | ko | en,fr,zh | fr,zh,ru | zh,ja,ko | all |
| Vertex | 0.19 | 0.18 | 0.19 | 0.22 | 0.20 | 0.19 | 0.21 | 0.37 | 0.37 | 0.38 | 0.51 |
| LLaVA | 0.22 | 0.29 | 0.28 | 0.29 | 0.36 | 0.40 | | 0.45 | 0.47 | 0.46 | 0.57 |
| | | | | | | | | en,fr | en,zh | fr,zh | |
| XM | 0.38 | 0.40 | 0.38 | 0.40 | 0.45 | 0.42 | 0.41 | 0.41 | 0.46 | 0.45 | 0.65 |

Table 5: Mean coverage across different linguistic measures. All values represented as percentages of the total metric range. 'en', 'de', etc. represent the metrics as calculated across a monolingual distribution. 'multi' represents metrics across a multilingual distribution of the same size. 'all' represents metrics across the entire multilingual distribution. Computed across Vertex captions. **C**

| | | Monolingual | | | | | | | | |
|---|---|---|---|---|---|---|---|---|---|---|
| | | en | de | fr | ru | zh | ja | ko | multi | all |
| Metric $M$ | Concreteness | 32.80 | 32.60 | 33.40 | 33.20 | 30.20 | 30.00 | 31.20 | 35.80 | 46.60 |
| | Analytic | 0.84 | 0.43 | 0.62 | 1.08 | 2.30 | 2.60 | 2.17 | 2.08 | 7.95 |
| | Clout | 4.94 | 5.05 | 5.4 | 7.14 | 6.92 | 6.49 | 5.56 | 11.16 | 27.15 |
| | Authentic | 23.21 | 22.8 | 21.68 | 23.07 | 23.51 | 25.01 | 21.67 | 39.5 | 76.47 |
| | Tone | 1.85 | 1.84 | 2.03 | 2.63 | 2.05 | 1.96 | 2.05 | 4.33 | 10.18 |
| | Color | 18.20 | 23.57 | 21.24 | 22.60 | 17.92 | 16.49 | 21.19 | 27.74 | 51.28 |

Table 6: Ground truth coverage ($|\mathbb{C} \cap \mathbb{G}|/|\mathbb{G}|$) increases when sampling multilingual captions. $\mathbb{C}$ refers to the caption concept set and $\mathbb{G}$ refers to the 'ground truth' Visual Genome concept set, per earlier notation. Relationships between monolingual and multilingual distributions are statistically significant with correction. As a baseline, $\mathbb{E}[|\mathbb{G}|] = 21.40$. $|\mathbb{C} \cap \mathbb{G}|$ (unnormalized intersection size, number of objects shared) and $|\mathbb{C}|$ are provided for reference. **D**

| | | | | Monolingual | | | | | | Multilingual | |
|---|---|---|---|---|---|---|---|---|---|---|---|
| | | en | de | fr | ru | zh | ja | ko | en,fr,zh | en,de,ja | de,fr,ru |
| Vertex | $|\mathbb{C} \cap \mathbb{G}|/|\mathbb{G}|$ | 0.183 | 0.171 | 0.177 | 0.185 | 0.170 | 0.162 | 0.156 | 0.200 | 0.197 | 0.225 |
| | $|\mathbb{C} \cap \mathbb{G}|$ | 3.49 | 3.28 | 3.38 | 3.53 | 3.23 | 3.08 | 2.96 | 3.84 | 3.79 | 3.64 |
| | $|\mathbb{C}|$ | 3.70 | 3.56 | 3.60 | 3.83 | 3.48 | 3.09 | 3.03 | 4.31 | 4.24 | 4.10 |
| LLaVA | $|\mathbb{C} \cap \mathbb{G}|/|\mathbb{G}|$ | 0.195 | 0.186 | 0.200 | 0.170 | 0.155 | | | 0.214 | | 0.215 |
| | $|\mathbb{C} \cap \mathbb{G}|$ | 3.79 | 3.60 | 3.90 | 3.29 | 3.01 | | | 4.21 | | 4.19 |
| | $|\mathbb{C}|$ | 4.58 | 5.32 | 5.66 | 4.83 | 4.78 | | | 6.28 | | 6.48 |

Appendix A. Multilingual captions identify undocumented objects at a higher rate than monolingual captions (on average, 10.9% vs 5.3%).

## 4 FINETUNING ON MULTILINGUAL VERSUS MONOLINGUAL DATA

Our analysis thus far highlights two important findings. First, vision-language datasets across different languages cover widely varying concepts. Second, models amplify these differences. In this section, we identify what these findings mean for when we finetune vision-language models with multilingual versus monolingual data.

**Method.** We finetune a pretrained GIT model (Wang et al., 2022) on 8 image captioning datasets. These finetuning datasets use the same set of 1.8k training images from the Crossmodal dataset. We automatically create these eight training datasets by translating captions generated from Vertex, LLaVa, and XM multilingual captions generated in the our analysis above into English.[3] and use the original Crossmodal's human generated data as finetuning data. The first seven training sets

---

[3] Translating all multilingual captions into English allows us to vary the description's semantic content without simultaneously changing the language and, thus, token distribution used by the model.

Table 7: SPICE F-scores when evaluating a model fine-tuned on the training set from the language on the left against the validation set from the language on the top. Vertex captions. For instance, a model fine-tuned on English Vertex captions obtains 0.219 SPICE score on Russian Vertex captions. 'multi' refers to an even split across all languages. Red indicates best performance on a split, yellow highlights model fine-tuned on 'multi'.

|  |  | **Evaluated on** | | | | | | | |
|  |  | **en** | **de** | **fr** | **ru** | **zh** | **ja** | **ko** | **multi** |
| | **en** | 0.271 | 0.225 | 0.229 | 0.219 | 0.218 | 0.229 | 0.232 | 0.230 |
| | **de** | 0.213 | 0.245 | 0.219 | 0.217 | 0.215 | 0.210 | 0.226 | 0.219 |
| | **fr** | 0.248 | 0.240 | 0.259 | 0.234 | 0.236 | 0.239 | 0.253 | 0.246 |
| **Fine-tuned on** | **ru** | 0.226 | 0.234 | 0.228 | 0.254 | 0.231 | 0.236 | 0.237 | 0.239 |
| | **zh** | 0.199 | 0.202 | 0.199 | 0.207 | 0.247 | 0.220 | 0.224 | 0.216 |
| | **ja** | 0.212 | 0.212 | 0.215 | 0.212 | 0.226 | 0.266 | 0.245 | 0.223 |
| | **ko** | 0.218 | 0.222 | 0.224 | 0.217 | 0.242 | 0.239 | 0.271 | 0.235 |
| | **multi** | 0.239 | 0.233 | 0.234 | 0.233 | 0.235 | 0.243 | 0.252 | 0.244 |

contain translations from a single language (i.e. we have a finetuning dataset for French, for German etc.). The eighth dataset is constructed with equal proportions of captions across all 7 languages. We choose GIT because of its simple architecture and robust performance across benchmarks.

We evaluate the finetuned models across a held out 1.8k images from the Crossmodal dataset. Validating each of the 8 fine-tuned models on each of the 8 holdout caption sets yields 64 evaluations. We use SPICE F1-score as our evaluation metric (Anderson et al., 2016), which measures conceptual overlap over low-level syntax adherence. We generate "ground truth" captions across the different languages using Google's Vertex commercial API, which again, we deem to be the highest quality given its commercial application and quality checks.

We interpret our results using the following logic. Let $\mathcal{L}(A; B)$ represent the error of a model fine-tuned on the data from language $B$ on the evaluation data from language $A$. If the scores across all evaluations $A$ are approximately equal, then the prevailing assumption that the content of captions across languages are approximately equal holds. On the other hand, if $\forall_{A \in \mathbb{L}} \forall_{B \in \mathbb{L}} \mathcal{L}(A; A) > \mathcal{L}(B; A)$, this is evidence against the prevailing assumption: models learn particular concepts when finetuned using data from a specific language, and performs better when evaluated against captions from that language.

**Results.** We find that a model fine-tuned on language $A$ performs best on language $A$ (Table 7), again against the prevailing assumption. We find the same results across models finetuned on data from LLaVA (Table 17), and from the human annotated Crossmodal dataset (Table 18). Moreover, finetuning multilingual data from Vertex yields a model which performs consistently well across all evaluation data compositions. Together, these suggest that models trained with multilingual data, representing diverse perceptual modes learn more about the visual world. One possible confounder is syntactic artifacts introduced during translation. For instance, text translated from German into English might have a unique syntactic structure which distinguishes it from text originally written in English. If this is the case, then it should be possible to identify translated text from one language versus another. To test this limitation, we embed all translated captions using a BERT-based model (Reimers & Gurevych, 2019). We fit a logistic regression model to predict a sample's original language from these features, and find near-random chance performance at 16.43%.

## 5 CONCLUSION

Similar to previous theories and empirical findings in social science that studies the subjectivity of human perception, we show differences between the content of captions from different languages in datasets, model behavior, and model training. In general, we turn from a constant and objective conception of the "ground truth" towards the human subject, which selectively attends to certain objects and qualities in the nearly-infinite field of raw visual features. Our work suggests that even the "objective concepts" within a scene are, at their root, observed by human subjects with particular perceptual tendencies, rather than independently existing.

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
