# A  FURTHER DISCUSSION

Table 8: A few examples of captions collected from the human study across English and Japanese speakers show differences in the observed content for each image. Japanese captions tend to include **more context** (e.g., background objects, added details). Samples are selected but representative of broader trends.

| | | | |
|---|---|---|---|
| | English | I.
II.
III. | Two very small boats on a river
toy boats in the water
A yellow boat and a red boat that appear to be models. |
| | Japanese | I.
II.
III. | Two boats on the water and **a building in the back**
close-up of a model of a boat and **people on the waterfront**
Two boats floating on the river and a **model of the town in the distance** |
| | English | I.
II.
III. | Luggage left unattended at a table.
Luggage lined up next to tables with jackets resting on the tables.
luggage sitting next to tables |
| | Japanese | I.
II.
III. | **A man sitting in a lobby** with lots of suitcases and bags
**A man is sitting in a room**, and there are several tables filled with luggage nearby.
**Man waiting** with a lot of luggage |
| | English | I.
II.
III. | Cat laying down in an arm chair.
A Siamese cat laying on its back on a couch next to a pillow.
A cat stretched out and upside down on a chair |
| | Japanese | I.
II.
III. | A cat stretches out on a blue chair and a pillow with **an embroidered owl** next to it.
A cat is relaxing next to a cushion with **a picture of an owl** on it.
Cat sitting on his back in an armchair with an **owl-patterned** cushion |

**From Object to (Human) Subject: Relativizing Ground Truth.**

Computer Vision often adopts an object-centric approach, in which a set of salient objects in visual scenes constitute a singular ground truth. This assumption is written into the very optimization objectives and metrics used in image captioning, which require the model to minimize the difference between predicted lexical units and the 'true' lexical units (BLEU Papineni et al. (2002), METEOR Banerjee & Lavie (2005)) or between predicted concepts and 'true' concepts (SPICE Anderson et al. (2016)). Even models such as DenseCap (Johnson et al. (2015a)) and dataset efforts such as Visual Genome (Krishna et al. (2016)) fail to cover *all* semantic information in a visual scene (see **D**). Neither is it the objective of image captioning to do so in the first place: the task of image captioning contains an irreducible subjective core.

Therefore, we must take into account patterns in subjects' attention towards certain object sets rather than simply upholding some finite object set as the ground truth. For instance, in our evaluations for the distillation/fine-tuning experiments **A**, we observe that image captioning models fine-tuned on the same images but captions following different content distributions behave differently when validated on differently distributed 'ground truth labels'.

We are not only suggesting that image captioning involves subjective processes such as emotion and aesthetic judgement (Mohamed et al. (2022)) or that language is fluid and can represent the same set of concepts in many different ways (Luong et al. (2021)). As we reiterate, *even the "objective concepts" within a scene are, at their root, observed by human subjects with particular perceptual tendencies.*

**Human-Centered AI and Social Biases**

Our work's turn from simply existing objects to the human subject fits within the broader research direction of human-centered AI (HAI). HAI strives to build and evaluate AI models in alignment with human values and behavior (Tahaei et al. (2023); Taylor et al. (2023)). Importantly, blindly building AI for "humans" in the abstract can substantively neglect populations which are not represented under that signifier, and possibly harm algorithmic equity and fairness goals (Skeem & Lowenkamp (2016); Buolamwini & Gebru (2018); Hershcovich et al. (2022)). For instance, Berg et al. (2012) argue that adopting a human-centric view of recognition is essential for developing practical image and video search applications: "*in response to an image search for "tree", returning an image with a tree that no person would ever mention is not desirable.*" However, failing to understand diversity in perception might lead an AI system designer to falsely claim that "no person" would mention some

tree, when in fact the persons who *would* have simply not been considered. The English-dominated language modeling landscape may not only introduce accessibility issues for non-English speakers but also possibly perceptual bias / hegemony (Kleinberg & Raghavan (2021)). In some ways, our contribution is similar to that of Caliskan et al. (2022) in showing that models (and datasets) encode and internalize social stereotypes (in their case) or content distributions (in our case).

**Reframing Language Accessibility**

Arguments in favor of directing attention towards multilingual NLP and CV are often made from an accessibility perspective of expanding information and technology access to individuals (Hu et al. (2020)) However, models empirically encounter the "curse of multilinguality" (Conneau et al. (2020)), in which increasing the number of languages a model handles degrades monolingual performance after a basic performance threshold. Therefore, expanding language accessibility is often seen as a trade-off against model performance and/or resource cost.

This appears to be even more so the case in CV than in NLP. If one accepts the previously discussed prevailing assumption in CV, then the visual data serves as an objective field of information which language – even with small deviations – must be constrained by. Therefore, many efforts to build vision-language systems for a non-English language $X$ train on translations from English to $X$ (Mishra et al. (2021); Chen et al. (2022)). Although these systems are usable in $X$, our investigation suggests that *they will still reflect an English content distribution* rather than an "organically $X$" content distribution.

Our work demonstrates a substantive system benefit to including multilingual data: increased conceptual richness, as measured both formally and empirically. Using organically produced multilingual data introduces shifts in the conceptual distribution which translation cannot. This is an "inversion of the curse of multilinguality": rather than expanding image captioning to multiple languages for accessibility goals at the expense of performance, incorporating the diverse content distribution may improve model robustness and grasp of diverse concepts.

**Linguistic Structure and Information**

What explains the observed differences between the content of monolingual distributions? There are some potential factors associated with ML landscape artifacts which cannot be eliminated, including the availability, quality, and context of data in different languages; the model training or data gathering procedures across languages; etc. Accounting for these factors, we can still examine patterns across languages. Particularly, we see the pattern that European languages (English, German, French, and to some extent Russian) are distributed more closely to each other than towards Asian languages (Chinese, Japanese, Korean), as demonstrated in Ⓐ in multilingual and multimodel intersections and elsewhere. Interestingly, besides the geographical proximity, the group similarities fall along shared linguistic typological (family) features. More research would be required to qualify their similarities in a more meaningful manner. However, the current observations aligns with the studies that show the linguistic, cultural, and historical influence the way we perceive the world (Boroditsky, 2006; Boroditsky et al., 2003; Lakoff, 1987) and that those influences have a tangible effect on how the people express through language their perceived world (Langacker, 1993; Talmy, 1988; Divjak, 2019).

Our experiments may point towards further dialogue on linguistic differences ('relativism') and its relevance in vision-language tasks. Unlike NLP tasks, which are often open-ended and not strictly tethered to reference material, vision-language tasks require bridging an open-ended language field with a constant visual reference field. Indeed, the very structure of language – grammar, convention, etc. – may incentivize the production of certain information over others.

In a strictly linguistic sense, translation approximates but does not directly represent meaning across languages, as word acquires its particular meaning within a language by its relations to other words within the language (de Saussure (1960)). Therefore, in this strict sense, the notion of a 'content distribution' without the language confounder is impossible. However, importantly, we are not as interested in being entirely faithful to the specific meaning determinations in a specific language as we are in understanding what can come out of representing uncommon or novel concepts in another language, even if they do not have exact correspondence with the original text.

# B  LIMITATIONS AND ETHICAL CONCERNS

While our paper sheds light on the relationship between language, culture, and visual perception, it is crucial to recognize and acknowledge its limitations as well as potential ethical ramifications.

## B.1  LIMITATIONS

**Overemphasis on Language.** While our study advocates for accounting of diverse sources of captioning, it relies on and possibly overemphasizes language as the major axis of diverse sourcing. Our experiments also found that captions generated by different models showed promise in increased information coverage and further research in this area can inform how to effectively source diverse concepts beyond using language as a proxy and a constraint. Multilingual sources are a sometimes readily available, but underutilized resource that not only contribute to inclusivity but has the potential to improve vision systems.

**Languages chosen.** We used only 7 languages in our study and hence our findings may not generalize beyond that. Languages were chosen to represent those spoken in "the West" and "the East" generally, but is still not representative of global linguistic diversity.

**Measurements.** Measuring semantic content is subjective, and hence most measures are merely proxies of actual information coverage. The choice of metrics we work with (i.e. scene graph, embedding, linguistic complexity) may also have inherent biases. There could be other ways to evaluate semantic content that were not considered in this study. Furthermore, the assumption that an increase in objects, relations, or attributes correlates with better understanding or improved perception can be questioned.

**Scale.** Following on from the preceding point, our findings are based on small-scale experiments. These findings may not scale to large scale models (e.g., CLIP), which are often considered the gold standard for image understanding. We resort to small-scale experiments for several reasons, including 1) focusing on breadth instead of depth, i.e. curating and analyzing human-annotated / model-generated image captions in seven languages. 2) focusing on the science underlying perceptual differences as documented in the literature of cross-cultural psychology and linguistics. In this paper, we study if the reported perceptual differences exist for computer vision tasks and how we might use them to improve information coverage in visual representations. Future studies may forego a controlled setup like ours to determine the validity of our findings on a larger scale, i.e. whether CLIP-scale models differ in their image understanding when trained with captions from different languages. Previous works have shown promise in this direction by showing how better / denser (albeit synthetic) captions can help with better image understanding. (Nguyen et al., 2023; Lai et al., 2023)

**Content Translation.** The paper relies on translating content (using GPT-4) from the original language to English in order to use the same robust tools that are generally only available in English. Moreover, such translation helps eliminate confounders such as scripts and their issues with tokenization. However, even with careful human studies to ensure acceptable quality, the translation process can be noisy, with the potential for errors to propagate. The translation process is also often considered to be more expensive than synthetic data, and so may not be economically viable to scale.

**Experimental design.** Our choices for selection of experiments, especially the model architecture and training process could influence some of the reported results. While we made effort to adhere to standard practices in computer vision, the ideas we propose and experiments we conduct are relatively new to the field, necessitating new forms of experiments that might be prone to our inherent biases. Moreeover, we were limited by the small number of currently accessible multilingual vision-language models to evaluate on and dense multilingual image captioning datasets to examine. With attention being increasingly being paid towards building multilingual vision-language models, such as PALI (Chen et al. (2022)), and high-quality multilingual vision-language datasets, we anticipate patterns to solidify and new problems to emerge.

**Cultural Interpretation & Relativism** The paper makes an assumption that some of linguistic differences are symptoms of cultural differences, which may oversimplify the intricate link between language and culture. Furthermore, several of the findings in the paper allude to language and culture influencing how people think and function. This proposition is often subject to heated debates, with

linguists and anthropologists opposing determinism (the stronger form) i.e. 'Language *shapes* the way we think' and supporting relativism (the lesser version) i.e. 'Language *affects* the way we think'. Linguistic relativism is frequently referred to as the 'Sapir-Whorf Hypothesis'. Boroditsky (2006)

## B.2 ETHICAL CONCERNS

**Cultural Essentialism.** By categorizing languages or cultures based on differences in image perception and captioning, there's a risk of essentializing or stereotyping cultures, suggesting that all members of a certain culture perceive things a certain way. This may also increase the risk of researchers seeing languages as merely objects for increasing the advancement of AI systems, rather than engaging with these languages and developing multilingual systems that work for people speaking a variety of languages.

## C TRANSLATION

We prompted GPT-4 (OpenAI (2023)) to translate text with: "Return the translation (and only the translation) of the following text from `[SRC_LANG]` into `[TGT_LANG]` exactly with all details: `[TEXT]`". We find that this prompt produces translations which are especially focused on representing the conceptual details of the original text.

Although language-specific meanings will inevitably lost in any translation between languages, we ensure that our English translations are as faithful as possible to the concepts expressed in the original language by conducting a human evaluation. We recruit 2-3 speakers for each of the six non-English languages (French, German, Russian, Chinese, Japanese, Korean), fluent in both the original image and English, from Prolific. Each subject evaluates 30 pairs of original and translated text. Of these 30 pairs, 10 are Vertex captions on Crossmodal images, 10 are LLaVA captions on Crossmodal images, and 10 are Vertex captions on Visual Genome images. This composition ensures wide coverage across image domains and caption format. Each translation evaluation has two parts. Firstly, subjects annotate the overall translation quality on a 1 to 5 scale, in which 1 is "entirely inaccurate", 2 is "some of the information is preserved", 3 is "only the most important information is preserved", 4 is "most of the information is preserved (the translation is adequate but not perfect)", and 5 is "entirely accurate". Secondly, subjects examine 11 general categories of concepts in natural visual scenes, provided by TIFA (Hu et al. (2023)): objects, animals/humans, attributes, activities, spatial relations, counting, food, materials, shapes, locations and colors. Subjects mark each category either as "Good" (the concept was present in the original text and faithfully represented in the translation), "Missed" (the concept was present in the original text but absent or not faithfully represented in the translation), or "N/A" (the concept was not present in the original text). Table 9 demonstrates that the translations are nearly entirely accurate, especially for European languages, and preserve nearly all of the salient content categories for understanding visual scenes.

Table 9: Human evaluations for translation quality using GPT-4 on multilingual captions. TIFA categories represent the mean proportion of non-N/A responses which are marked "Good" (as opposed to "Missed").

|  | Metric | de | fr | ru | zh | ja | ko |
|---|---|---|---|---|---|---|---|
| Quality Ratings | Mean | 4.95 | 4.76 | 4.82 | 4.63 | 4.48 | 4.48 |
|  | Median | 5.00 | 5.00 | 5.00 | 5.00 | 5.00 | 5.00 |
|  | 25th Percentile | 5.00 | 5.00 | 5.00 | 4.00 | 4.00 | 4.00 |
| TIFA Categories | Objects | 1.00 | 0.99 | 1.0 | 0.97 | 0.98 | 0.90 |
|  | Animals/Humans | 1.00 | 1.00 | 1.00 | 1.00 | 1.00 | 1.00 |
|  | Attributes | 1.00 | 0.89 | 1.00 | 0.93 | 1.00 | 1.00 |
|  | Activities | 1.00 | 1.00 | 1.00 | 0.98 | 0.91 | 0.96 |
|  | Spatial Relations | 1.00 | 1.00 | 1.00 | 0.92 | 0.94 | 0.96 |
|  | Counting | 1.00 | 1.00 | 1.00 | 0.99 | 1.00 | 0.90 |
|  | Food | 1.00 | 1.00 | 1.00 | 1.00 | 1.00 | 1.00 |
|  | Material | 1.00 | 1.00 | 1.00 | 1.00 | 1.00 | 1.00 |
|  | Shape | 1.00 | 1.00 | 1.00 | 1.00 | 1.00 | 1.00 |
|  | Location | 1.00 | 0.98 | 1.00 | 0.98 | 0.96 | 0.89 |
|  | Color | 1.00 | 0.96 | 1.00 | 0.97 | 1.00 | 1.00 |

Annotators were allowed to provide free-text explanations for areas in which the translation was inadequate. We provide a random sampling of comments, to provide a holistic idea of the translation weaknesses.

Table 10: Example comments providing corrections to translations.

| Comment |
| --- |
| ↪ Should use "above" instead of "on" |
| ↪ More appropriate to use "memories" instead of "impressions" |
| ↪ should be 'small' balls (remove 'round', add 'small') |
| ↪ Particle suggests that numbers are written "using" sheet of paper, not "on" it. |
| ↪ "On the side" is translated as "next to". |
| ↪ "toile d' araignée" can be directly translated to "cobweb", it's a bit excessive to say spider & spider web in the translation |

## D   Miscellaneous Experiment Details

### D.1   Probing Multilingual Capabilities in LLaVA

Models like LLaVA which are trained/fine-tuned/aligned with English data but which include multilingual LLM components can retain some of these multilingual capabilities. In order to request LLaVA generate captions in a target language, we change the prompt at all levels to correspond to that language language, displayed in Table 11.

Table 11: Prompt information for probing multilingual behavior in LLaVA.

| Prompt Type | Language | Prompt |
| --- | --- | --- |
| **Roles** | English | (user, assistant) |
| | French | (utilisateur, assistant) |
| | German | (Benutzer, Assistent) |
| | ... | ... |
| **System** | English | A conversation between a user and an LLM-based AI assistant. The assistant gives helpful and honest answers. |
| | French | Une conversation entre un utilisateur et un assistant IA basé sur LLM. L'assistant donne des réponses utiles et honnêtes. |
| | German | Ein Gespräch zwischen einem Benutzer und einem auf LLM basierenden KI-Assistenten. Der Assistent gibt hilfreiche und ehrliche Antworten. |
| | ... | ... |
| **User Prompt** | English | What is in this image? Answer in English. |
| | French | Qu'est-ce qu'il y a dans cette image? Répondez en français. |
| | German | Was ist auf diesem Bild? Antwort auf Deutsch. |
| | ... | ... |

We unsuccessfully attempted to probe multilingual capabilities in other models in a similar fashion, such as the BLIP family (Li et al. (2023a)).

### D.2   Image Captioning User Study

We recruited 10 English speakers from the US and 10 Japanese speakers from Japan from Prolific. The instructions given to them are presented in Figures 4a and 4b. A sample of the produced captions is given in Table 8.

### D.3   Supplementary Normalized Continual Graph Expansion Charts

The continual scene graph expansion as measured by number of objects, attributes, and relations in Figure 3. This pattern also holds when measuring by object-standardized measures: number of attributes per object (Figure 5a) and number of relations per object (Figure 5b).

### D.4   Pairwise Monolingual Scene Graph Intersections 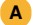

Table 12 displays the sizes of intersections between monolingual scene graphs as measured by the number of objects and relations, using the formula $M(A) + M(B) - M(A \cup B) = M(A \cap B)$.

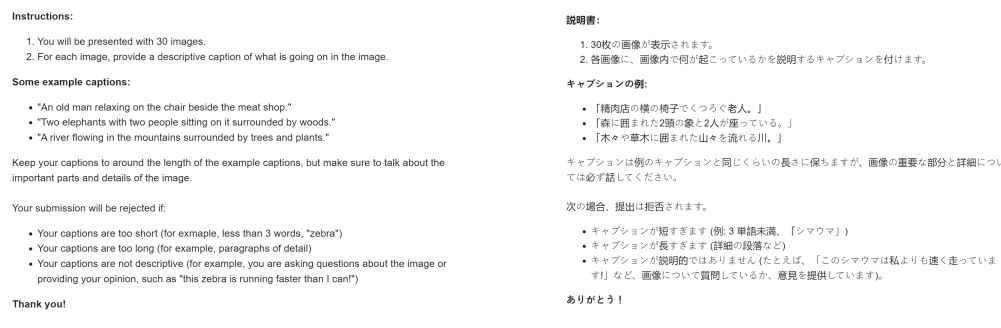

(a) English instructions.

(b) Japanese instructions.

Figure 4: Instructions and examples presented to human evaluation participants for image captioning.

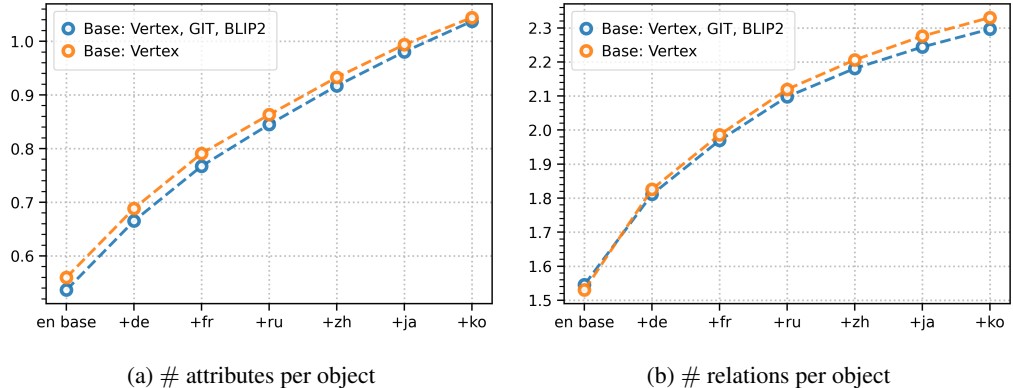

(a) # attributes per object

(b) # relations per object

Figure 5: Growth in scene graphs by successively unioning with monolingual scene graphs in each language.

Table 12: Sizes of intersections between monolingual unioned scene graphs, listed in the form "number of objects / number of relations". Sizes listed along the diagonal arc of monolingual graphs and can be used for reference.

|      | en          | de          | fr          | ru          | zh          | ja          | ko          |
|------|-------------|-------------|-------------|-------------|-------------|-------------|-------------|
| en   | 3.65 / 2.96 | 2.34 / 1.20 | 2.41 / 1.26 | 2.39 / 1.24 | 2.15 / 0.97 | 2.04 / 0.91 | 2.02 / 0.89 |
| de   |             | 3.51 / 2.83 | 2.37 / 1.24 | 2.47 / 1.32 | 2.14 / 0.98 | 2.04 / 0.91 | 2.06 / 0.94 |
| fr   |             |             | 3.60 / 2.89 | 2.44 / 1.27 | 2.16 / 0.97 | 2.07 / 0.91 | 2.09 / 0.95 |
| ru   |             |             |             | 3.86 / 3.2  | 2.25 / 1.06 | 2.13 / 0.98 | 2.12 / 0.96 |
| zh   |             |             |             |             | 3.46 / 2.68 | 2.08 / 0.95 | 2.04 / 0.92 |
| ja   |             |             |             |             |             | 3.13 / 2.37 | 2.10 / 1.02 |
| ko   |             |             |             |             |             |             | 3.18 / 2.47 |

## D.5    REPRESENTATIONAL DIVERSITY FOR CROSSMODAL  B

The Crossmodal dataset provides at least two captions per language for each image. We compute the mean coverage as defined in  B  between pairwise languages.

## D.6    ADDITIONAL LINGUISTIC METRICS  C

Table 14 shows linguistic metrics in the same format as Table 5 for LLaVA captions, and Table 15 for XM captions. Observe that, in most cases, resampling from a multilingual distribution increases the mean coverage of each linguistic measure.

Table 13: Language-pairwise mean coverage of embedding space, computed across the Crossmodal dataset.

|    | en | de | fr | ru | zh | ja | ko |
|----|------|------|------|------|------|------|------|
| en | 0.38 | 0.43 | 0.41 | 0.44 | 0.46 | 0.44 | 0.45 |
| de | 0.43 | 0.40 | 0.40 | 0.42 | 0.46 | 0.42 | 0.44 |
| fr | 0.41 | 0.40 | 0.38 | 0.42 | 0.45 | 0.42 | 0.42 |
| ru | 0.44 | 0.42 | 0.42 | 0.40 | 0.47 | 0.44 | 0.45 |
| zh | 0.46 | 0.46 | 0.45 | 0.47 | 0.45 | 0.46 | 0.46 |
| ja | 0.44 | 0.42 | 0.42 | 0.44 | 0.46 | 0.42 | 0.42 |
| ko | 0.45 | 0.44 | 0.42 | 0.45 | 0.46 | 0.42 | 0.41 |

Table 14: Mean coverage across different linguistic measures, as in Table 5. Computed across LLaVA captions.

| | | Monolingual | | | | | | | |
|---|---|---|---|---|---|---|---|---|---|
| | | en | de | fr | ru | zh | ja | multi | all |
| | Concreteness | 2.17 | 2.44 | 2.53 | 2.27 | 2.33 | 2.21 | 2.57 | 3.19 |
| | Analytic | 2.85 | 4.64 | 14.72 | 7.53 | 23.05 | 20.86 | 18.09 | 44.48 |
| Metric $M$ | Clout | 14.19 | 19.96 | 15.81 | 23.15 | 21.3 | 29.57 | 29.14 | 67.1 |
| | Authentic | 35.97 | 38.01 | 34.33 | 37.20 | 44.63 | 42.92 | 52.59 | 90.58 |
| | Tone | 6.79 | 9.53 | 16.64 | 16.48 | 11.56 | 9.55 | 16.56 | 45.97 |
| | Color | 18.8 | 26.5 | 17.46 | 12.63 | 17.98 | 13.55 | 24.97 | 69.21 |

Table 15: Mean coverage across different linguistic measures, as in Table 5. Computed across XM captions.

| | | Monolingual | | | | | | | | |
|---|---|---|---|---|---|---|---|---|---|---|
| | | en | de | fr | ru | zh | ja | ko | multi | all |
| | Concreteness | 36.06 | 44.74 | 42.16 | 40.51 | 45.13 | 44.90 | 42.70 | 43.28 | 59.65 |
| | Analytic | 1.88 | 2.31 | 1.62 | 0.76 | 5.02 | 5.28 | 5.13 | 4.66 | 16.56 |
| Metric $M$ | Clout | 9.66 | 14.77 | 15.00 | 14.14 | 13.32 | 13.10 | 12.24 | 19.66 | 47.16 |
| | Authentic | 33.21 | 40.09 | 42.22 | 32.35 | 34.02 | 33.83 | 35.28 | 51.87 | 89.05 |
| | Tone | 8.62 | 12.16 | 9.74 | 10.90 | 10.59 | 9.18 | 9.07 | 13.88 | 40.24 |
| | Color | 11.73 | 34.59 | 24.90 | 27.74 | 48.98 | 36.30 | 16.37 | 40.28 | 94.01 |

## D.7  EXAMPLES, MULTILINGUAL DISTRIBUTIONS EXCEED VISUAL GENOME  D

Table 16 provides examples of Visual Genome images in which multilingual scene graphs mention objects which are not even documented in Visual Genome, despite it being a very densely annotated dataset.

## D.8  FINETUNING/DISTILLATION EXPERIMENTS

Table 17 and Table 18 display evaluations of models finetuned on one language composition against test data of another language composition, using LLaVA and XM captions, respectively. Observe that the patterns are similar to those in Table 7, albeit weaker as expected due to less stability in LLaVA and a dataset like XM.

## D.9  GPT-4 SAMPLES

We provide a few more examples of captions in English and Chinese in Table 19 from the (at the time of this paper) recently released GPT-4 vision model to gesture at possible trends in other models.

## D.10  CAPTION SAMPLES

We provide several image samples and their captions in English, German, French, Russian, Chinese, Japanese, and Korean by the Vertex API for reference in Table 20.

Table 16: Examples in which multilingual distributions identify visual features which are not documented in the Visual Genome dataset. Rightmost column indicates objects mentioned in multilingual scene graphs but which appear not to be covered in the Visual Genome object list.

| Image | Visual Genome, Documented Objects | + Multiling. Scene Graph Objects |
|---|---|---|
|  | woman, sign, man, bag, license plate, car, person, leg, satchel | umbrella, sandwich restaurant, street, rain |
|  | pane, door, shirt, box, cd, glass pane | wheelchair |
|  | woman, key, notes, page, keyboard, pencil case, laptop, student | table |
|  | leaves, sign, sky, cloud, trees, roof, train, steam cloud, ground, lamp, green leaves, cables, pole, tracks, locomotive, train car, tree, steeples, gravel, steam, bush, door, wheel | number, logo, inscription |
|  | tray, writing, cloth, stove door, light, oven back, bird necklace, mitt, shirt, apron, stove, burner, strings, aprontop, towel, board, pizza, shortsleeveshirt, menu, woman, necklace, pizzas, pan, oven, sheet | chalkboard |
|  | standing person, car, city, young girl, tree, person, boy, pink clothes, street | parking lot, parking meter |
|  | leaves, sign, car, light, yield sign, tree, picture | road, road intersection, arrow, street sign |
|  | giraffe tail, spot, rock, giraffe, rocks, grass | bird (left of image) |

Table 17: Evaluations for models fine-tuned on LLaVA captions. Generally speaking, a model fine-tuning on a particular language performs best on that language.

| | | Evaluated on | | | | | |
|---|---|---|---|---|---|---|---|
| | | en | de | fr | ru | zh | multi |
| **Fine-tuned on** | **en** | 0.271 | 0.225 | 0.229 | 0.219 | 0.218 | 0.230 |
| | **de** | 0.213 | 0.245 | 0.219 | 0.217 | 0.215 | 0.219 |
| | **fr** | 0.248 | 0.240 | 0.259 | 0.234 | 0.236 | 0.246 |
| | **ru** | 0.226 | 0.234 | 0.228 | 0.254 | 0.231 | 0.239 |
| | **zh** | 0.199 | 0.202 | 0.199 | 0.207 | 0.247 | 0.216 |
| | **multi** | 0.239 | 0.233 | 0.234 | 0.233 | 0.235 | 0.244 |

Table 18: Evaluations for models fine-tuned on XM captions. Generally speaking, a model fine-tuning on a particular language performs best on that language.

| | | Evaluated on | | | | | | | |
|---|---|---|---|---|---|---|---|---|---|
| | | en | de | fr | ru | zh | ja | ko | multi |
| **Fine-tuned on** | **en** | 0.254 | 0.124 | 0.1421 | 0.120 | 0.114 | 0.129 | 0.130 | 0.148 |
| | **de** | 0.158 | 0.153 | 0.152 | 0.143 | 0.124 | 0.140 | 0.146 | 0.149 |
| | **fr** | 0.182 | 0.142 | 0.181 | 0.143 | 0.130 | 0.146 | 0.150 | 0.154 |
| | **ru** | 0.172 | 0.136 | 0.152 | 0.159 | 0.125 | 0.137 | 0.142 | 0.148 |
| | **zh** | 0.144 | 0.116 | 0.129 | 0.120 | 0.124 | 0.130 | 0.142 | 0.130 |
| | **ja** | 0.144 | 0.128 | 0.137 | 0.125 | 0.124 | 0.154 | 0.144 | 0.135 |
| | **ko** | 0.151 | 0.116 | 0.131 | 0.116 | 0.115 | 0.134 | 0.159 | 0.134 |
| | **multi** | 0.179 | 0.140 | 0.153 | 0.145 | 0.131 | 0.149 | 0.151 | 0.151 |

Table 19: Examples of GPT-4 captions on Visual Genome images.

| Image | English | Chinese (Translated) |
|---|---|---|
| 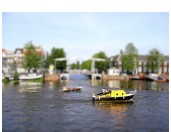 | The image shows a body of water, likely a canal, with boats. In the foreground, there's a boat with a yellow top and what appears to be a person inside, rendered with a tilt-shift photography effect that makes the scene look miniature. The background features architecture indicative of a European city, possibly Dutch, given the style of the bridge and houses. The tilt-shift effect creates a shallow depth of field, blurring the background and enhancing the miniature appearance of the scene. | This image shows a canal scene that appears to be somewhere in Europe, specifically Amsterdam, the Netherlands. The picture shows a canal lined with houses and trees, with a bridge in the background. In the center of the canal, there is a small yellow and black boat, and there seems to be someone on board. The image uses a tilt-shift effect to make the image look like a miniature model. The surrounding scenery is reflected on the water, and the whole scene is filled with a tranquil and vivid atmosphere. |
| 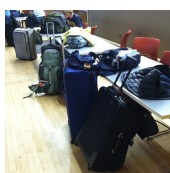 | The image shows a room with wooden flooring where several bags and pieces of luggage are placed on the floor and on top of a table. There are different types of bags including backpacks, a duffel bag, and wheeled suitcases. Some clothing items and possibly a coat are also visible on the table. There appears to be a person seated in the background, partially obscured by the table, wearing a blue top. The setting suggests this could be a waiting area, possibly at an airport, train station, or a similar transit location. | This image shows an indoor scene with several pieces of luggage placed next to a long table and on the floor. At least four pieces of luggage can be seen: a patterned hard-shell suitcase with spinner wheels, a green backpack, a dark blue hard-shell suitcase with spinner wheels, and a large black suitcase. On the table, scattered are several pieces of clothing, including a blue coat, a black jacket and some smaller items such as a wallet and a book. In the background there are some red chairs and a man wearing a yellow shirt is sitting, but most of the man's body is obscured by the table. The entire scene gives the impression of waiting or a temporary stop, perhaps at a resting place during travel. |
| 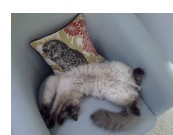 | The image shows a light-colored cat with darker points (likely a Siamese or a similar breed) in a relaxed, upside-down position on a blue chair. Behind the cat is a decorative pillow with a pattern that includes a red flower and what appears to be an owl. The scene looks cozy and domestic, with the cat appearing quite comfortable and at ease, suggesting it's in a familiar home environment. | This image features a long-haired Siamese cat lying in a very relaxed and comfortable position on a blue modern style armchair. The cat's body seems to be curled up softly in the curved part of the chair, with its head hanging upside down and its limbs dangling in the air, showing its relaxation and trust. The cat's coat color is typical of Siamese cats, with a light body and dark face, ears, feet and tail. Behind the chair there is a decorative pillow with a bird print, its red and brown pattern contrasting with the colors of the cat and the chair. The overall scene looks warm and peaceful. |
| 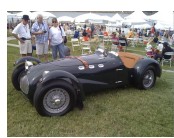 | The image displays a vintage black roadster with a tan interior. It has the classic design features of early to mid-20th-century sports cars, such as the rounded bodywork, prominent front wheel arches, wire-spoked wheels, and the absence of a fixed roof. The vehicle is parked on a grassy field with various people and white tents in the background, suggesting this could be at a car show or an outdoor event. There are three individuals standing near the car, two men and one woman, all casually dressed, possibly spectators or attendees of the event. | This picture shows a black retro-style sports car with smooth body lines and spoked wheels. The car is parked on a grassy field with an outdoor event scene in the background, complete with tents and people sitting in folding chairs. There were three people standing next to the car and it looked like they were discussing the car. One of them is wearing a camera, and the other is holding a document, perhaps a brochure for the event. This could be an antique car show or a gathering of car enthusiasts. |
| 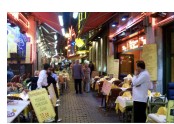 | The image shows a bustling European-style alley lined with restaurants and cafes. There are outdoor seating areas with yellow tablecloths, and large menus displayed on stands. Neon signs illuminate the area, suggesting a lively dining atmosphere. People are walking down the alley, some are seated at tables, and a man in the foreground is on a phone call. The cobblestone pavement and architectural details suggest this might be a historic or tourist area. The mood is convivial, typical of a dining district in the evening. | This picture shows a busy restaurant street. The photo shows outdoor seating at several restaurants, with tables and chairs lined with yellow tablecloths on both sides of the street. Some menu boards are placed in front of restaurants, showing prices and gastronomic options, such as "Moules Marinière". There were several pedestrians on the street, including a middle-aged couple taking a walk. On the right, a man in a white shirt is talking on the phone. The street is paved with cobblestones, and the buildings on both sides are decorated with neon lights and signboards, creating a cheerful and warm atmosphere. Overall, this photo captures the bustling dining scene of the city at night. |
| 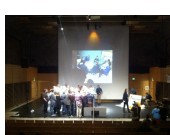 | The image depicts an indoor event in an auditorium or lecture hall setting. There's a crowd of people gathered on a stage; some of them are standing, while others are sitting on chairs. In the foreground, a few individuals appear to be setting up or adjusting a large sign or display that reads "PELOTON". A projection screen is visible in the background, displaying what seems to be a photo of a meeting or discussion. The lighting suggests the setting could be a conference, workshop, or a corporate event. The venue has tiered seating, and the architecture includes wooden paneling on the walls, enhancing the acoustics, which is typical for halls designed for speeches or performances. | This image shows an indoor gathering scene. There was a group of people gathered at the front desk, looking like they were having some kind of discussion or preparation. The background of the venue was a large screen showing another scene of what looked like several people sitting around a table. There is a sign with the word "Peloton" in the center of the front desk, which may indicate that this event is related to the Peloton company. There is a person standing at a microphone on the left side of the venue, possibly the host or speaker. Others were spread out in different areas of the venue, some sitting and some standing. Overall, this photo gives the feel of a formal party or event. |

Table 20: Sample captions from Vertex API in different languages on Crossmodal dataset images (translated into English).

| Image | Language | Translated Caption |
|---|---|---|
|  | en | A large body of water with a cliff in the background. |
| | de | A large rock in the middle of the ocean. |
| | fr | A mountain rises above the ocean. |
| | ru | Rocks on the ocean shore and their reflection in the water. |
| | zh | There is a group of rocks on the water surface. |
| | ja | There is a big rock in the middle of the sea. |
| | ko | Two rocks standing on the sea. |
|  | en | A bunch of bananas are stacked on top of each other. |
| | de | A bunch of bananas with black spots on them. |
| | fr | A close-up of a bunch of yellow bananas. |
| | ru | A bunch of bananas with black spots on the skin. |
| | zh | A close-up of a bunch of yellow bananas. |
| | ja | Close-up of a bunch of bananas. |
| | ko | A bunch of bananas is lying on the floor. |
|  | en | A man wearing glasses and a blue shirt smiles in front of a river. |
| | de | A man with glasses smiles in front of a river. |
| | fr | A man smiles next to a river and a forest. |
| | ru | A man in glasses and a blue shirt is smiling in front of the forest. |
| | zh | A man wearing glasses is standing by the river smiling. |
| | ja | A man wearing glasses is smiling at the camera in a blue shirt. |
| | ko | A man wearing glasses and a blue shirt is smiling towards the camera. |
|  | en | A rusty railing with a drawing of two people on it. |
| | de | A destroyed building with a rusty metal railing in front of it. |
| | fr | An abandoned building with stairs and a rusty fence. |
| | ru | Concrete wall with a rusty handrail and stairs. |
| | zh | On the wall, there is a silhouette of a man and a woman. |
| | ja | Stairs in front of a building with a rusted handrail. |
| | ko | Stairs in front of the building with a metal fence. |
|  | en | A man and a woman are walking down a street and the woman is looking at her phone. |
| | de | A man and a woman are standing on a street and looking at their mobile phone. |
| | fr | A man carrying a backpack with a european union flag on it. |
| | ru | A man with a backpack is standing next to a woman standing on the street. |
| | zh | A man with a backpack stands next to a woman. |
| | ja | A man and a woman are walking down the street in the city. |
| | ko | A woman is standing next to a man. |
|  | en | The inside of a temple with chinese writing on the ceiling. |
| | de | A chinese temple with a statue in the middle. |
| | fr | The interior of a temple with a red wooden ceiling. |
| | ru | A temple with an image of buddha in the center. |
| | zh | A large red and gold building, inside which there is a buddha statue. |
| | ja | Inside the temple, there is a buddha statue. |
| | ko | The interior of the temple is decorated in red and gold colors. |
|  | en | A park with trees and flowers and a building in the background. |
| | de | A park with trees and flowers and a building in the background. |
| | fr | A group of trees in a park with a building in the background. |
| | ru | A park with trees and flowers and a building in the background. |
| | zh | There is a bunch of blue and white flowers on the grass in the park. |
| | ja | Blue flowers are blooming in the middle of the park. |
| | ko | A blue flower bed in a park with a building in the background. |
|  | en | A close up of a plate of food with chopsticks on a table. |
| | de | Three rice rolls with sauce on a white plate. |
| | fr | A close-up of a roll of food on a white plate. |
| | ru | Dish on a white plate with sauce and sticks. |
| | zh | A plate of food with sauce and chopsticks. |
| | ja | Three roll breads on a white plate with sauce poured on them. |
| | ko | A food plate topped with sauce and chopsticks. |