# OpenReview forum: "Cultural and Linguistic Diversity Improves Visual Representations"
_ICLR.cc/2024/Conference — Submitted to ICLR 2024_

### Official Review · Reviewer_VFGo · 2023-10-29

**Soundness:** 2 fair
**Presentation:** 1 poor
**Contribution:** 2 fair
**Rating:** 3
**Confidence:** 5

**Summary:**

This paper presents an analysis of the properties of multilingual image captions produced by either humans or image captioning models. The analysis is based on the XM3600 dataset, and a collection of recently proposed models including LLaVA, BLIP2, GIT, and the Google Vertex API. Each of these models is used to generate image captions in seven languages, after which the original non-English output is translated into English using GPT-4. The analysis is broken down into looking at the complexity of the scene graph, representational diversity, linguistic diversity, and ground-truth coverage. The main claim, across all of these analyses, is that multilingual captions -- translated into English -- increase the diversity of the objects, relations, and linguistic complexity, compared to English-only captions.

**Strengths:**

The idea of determining the differences and similarities between image captions in different languages is interesting. It would be useful for the community to know, from an empirical perspective, if there are interesting or noteworthy differences in the captions across sets of languages. An effort to automate this using off-the-shelf language processing tools is a worthwhile endeavour.

**Weaknesses:**

The use of machine translation systems to have a working set of data that is only in English is a major weakness of the work. There is no explanation of possibly failed attempts to use non-English parsers or text embedding. Given the lack of meaningful discussion on this part, it is unclear whether this could infact work using the non-English languages directly.

The main claims of the paper are about how different languages can be shown to express things differently. However, the only example of the differences between English and non-English text, shown in Table 16 in the Appendix, has two dubious examples of the type of diversity found in translated versions of other languages. From these examples and the comments, it looks like the non-English captions include entities that are synonymous with other English entities (see Question 2.)

The related work section neglects to discuss many years of prior work in multilingual multimodal research, including but not limited to Wang et al. CVPR 2019,  Liu et al. EMNLP 2021, Srinivasan et al. SIGIR 2021, amongst many others. The authors are encouraged to consult a recent survey on this topic: Erdem et al. JAIR 2022.

Wang, Xin, et al. "Vatex: A large-scale, high-quality multilingual dataset for video-and-language research." Proceedings of the IEEE/CVF International Conference on Computer Vision. 2019

Liu, Fangyu, et al. "Visually Grounded Reasoning across Languages and Cultures." Proceedings of the 2021 Conference on Empirical Methods in Natural Language Processing. 2021.

Srinivasan, Krishna, et al. "Wit: Wikipedia-based image text dataset for multimodal multilingual machine learning." Proceedings of the 44th International ACM SIGIR Conference on Research and Development in Information Retrieval. 2021.

Erdem, Erkut, et al. "Neural natural language generation: A survey on multilinguality, multimodality, controllability and learning." Journal of Artificial Intelligence Research 73 (2022): 1131-1207.

**Questions:**

1. What efforts did you make to conduct your analysis in the original languages? In Section 3.1, you write "we require the consistent application of linguistic tools, such as parsers, which are more mature for English than other languages".
2. What are the main types of differences driving the differences in different non-English translated text? What types of objects or entities are being created? The examples in Table 16 are not so convincing to me: "alleyway" can be synonymous with "street", while "screen" can be synonymous with "monitor".
3. Why did you use the GPT-4 API instead of a dedicate machine translation model, such as M2M-100 or NLLB? I could not find any justification for not using a translation model in the paper.
4. Why are the final conclusions reliable when you have used a machine translation tool to convert the non-English text into English text?
5. What are the actual benefits of the Google Vertex API? The paper describes that the "outputs reflect established standards and have gone through quality checks" but I don't understand what that is supposed to mean.
6. How much money was spent on API calls? This would be useful for future researchers to know in attempting to reproduce your results.

---

> ### Author Response · Authors · 2023-11-21
>
> We are grateful to the reviewer for taking the time to provide valuable feedback on our work. We address below the questions raised in the review:
>
> 1. “What efforts did you make to conduct your analysis in the original languages?”
>
> The goal of our paper is to evaluate differences in caption content. We use a range of metrics to measure qualities of content. Some of these are implemented only in English. For instance, the LIWC metrics [1] are hand-crafted by linguists and therefore difficult to reproduce in other languages. In these cases, it is not possible to use non-English data. Other metrics have multilingual implementations, but they are not language agnostic. For instance, even advanced multilingual embeddings have been shown to encode language-specific information, especially typology [2, 3, 4, 5]. Implementations of linguistic parsers across languages vary widely in implementation and performance because their design depends on features of that language [6, 7]. Neural-based approaches for parsing multiple languages with one model usually rely on translation anyway [8]. In sum, we found analysis in original languages to be technically infeasible and introducing too many confounders, compared to translating to English.
>
> 2. “What are the main types of differences driving the differences in different non-English translated text?”
>
> We agree that the examples in Table 16 on ground truth coverage in Visual Genome are not so strong. We have since updated that figure with several more representative examples. We do suggest looking at Figure 2, which gives some examples of English and non-English captions and demonstrates that captions in different languages are mentioning different objects.
>
> It was hard for us to empirically validate clear, human-interpretable patterns between languages, although we do have many hypotheses we are testing for future work. For instance, some languages seem to mention objects in the background with greater frequency or mention certain types of attributes more often, which appears to align with existing work in cross-cultural psychology. See additional examples we have added in Tables 16, 19, 20 for more samples.
>
> The main contribution of our work is to show substantive differences between the content of captions in different languages across many metrics, and we hope future work can investigate explanations.
>
> 3. “Why did you use the GPT-4 API instead of a dedicate machine translation model, such as M2M-100 or NLLB?”
>
> M2M-100 and NLLB are both good models, but we found that we can specify to GPT-4 that we want our translations to preserve all factual details and to be exact. Machine translation models can optimize over many metrics, including factual correctness, but also including style, tone, coherence, and fluency, which can be in conflict with factual correctness [9]. For our purposes, factual correctness takes ultimate precedence over other factors.
>
> Although GPT-4 was released recently and we could not find published evaluations of GPT-4 on translation directly, several works have demonstrated GPT-4’s competence and even superior performance in some cases on multilingual tasks [10, 11, 12].

---

> ### Author Response · Authors · 2023-11-21
>
> 4. “Why are the final conclusions reliable when you have used a machine translation tool to convert the non-English text into English text?”
>
> We run a user study and show that native speakers find that the content of languages is preserved across translation (see Appendix A.2). We ensure that objects, their attributes, and relations between them are preserved. Consider the example of a bird in the background. A caption which mentions a bird will result in a translation which has a bird, and a caption which does not will result in a translation which does not have a bird. Therefore, we believe the signal we are measuring (the presence of objects, attributes, relations) is not likely to be sensitive to artifacts introduced during translation.
>
> 5. “What are the actual benefits of the Google Vertex API?”
>
> Google Vertex API represents one popular commercial solution to multilingual image captioning. By "outputs reflect established standards and have gone through quality checks", we simply mean that Google has done extensive testing to ensure the model’s behavior [13] and that it is a worthy benchmark to analyze.
>
> While we were running the experiments, we struggled to find comparable, high-quality, and open-source multilingual image captioning models. Since then, we have gained access to GPT-4V and added several examples in the appendix – see Table 19 in Appendix D.9.
>
> 6. “How much money was spent on API calls?”
>
> For the Vertex API, we made [3,600 images × 7 languages × 3 captions each = 75,600] calls and spent [75600 × 0.0015 = 113.4] for CrossModal images; and made [2,000 images × 7 languages × 3 captions each = 42,000] calls and [42,000 × 0.0015 = 63.] for Visual Genome images. Translation costs with GPT-4 total approximately \$597 to translate over 260k captions from non-English languages to English. We will also release all of our datasets upon acceptance for reproducibility.

---

> ### Author Response · Authors · 2023-11-21
>
> In response to each of the weaknesses raised:
>
> 1. Conducting analysis only in English.
>
> We believe that conducting our analysis in English is actually a strength. Suppose running analysis entirely in captions’ original languages were possible (which, as discussed in response to Question 1, it is not). Even then, a strong objection to any results we might produce is that language acts as a confounder. For instance, in our fine-tuning experiments, it might be that observed differences are simply due to differences in pretraining material, data availability, tokenization procedures, etc.
>
> Given that we have measured that relevant content (objects, attributes, relations) are preserved over translation (response to Question 4), conducting the analysis in English allows us to make a strong claim about content differences in captions from different languages.
>
> 2. Weak examples of differences between English and non-English.
>
> In addition to Table 2, we have added more examples of English and non-English captions in Table 20 (Appendix D.10). Additionally, as mentioned before, we have added GPT-4 examples in Table 19, Appendix D.9.
>
> 3. Not discussing previous work in multilingual multimodal modeling.
>
> We appreciate your recommended literature on multilingual multimodal research. We have added these as references in the paper. However, we believe that the suggested work is complementary yet ultimately different from our contributions.
>
> The suggested previous work in multilingual vision covered by [14] has focused on expanding vision capabilities from monolingual (English) contexts outwards to other languages (“getting models in different languages to see the same things”). We investigate how vision models built from different languages might behave differently (“observing how models in different languages see different things”).
>
> To this end, the suggested VaTeX [15] and WIT [16] datasets are helpful, and be good bases for future work. The suggested Liu & Bugliarello et al. paper [17] focus more on differences in knowledge of cultural objects and practices, e.g., dresses or foods particular to some culture. While this is parallel to our work in that it also studies differences in model behavior across human groups, we focus on investigating differences in perception of generally universal objects, e.g. birds, trees, and cars.
>
> If interested, we further discuss our unique contributions in Appendix A.1 (Discussion).
>
>
> We hope this helps clarify! Please let us know if there are any other comments or questions which may arise. If we addressed your concerns, we would appreciate an increase in your score.
>
> – the authors

---

> > ### Author Response · Authors · 2023-11-21
> >
> > References
> >
> > [1] https://www.liwc.app/
> >
> > [2] Libovický, J., Rosa, R., & Fraser, A.M. (2020). On the Language Neutrality of Pre-trained Multilingual Representations. Findings of EMNLP.
> >
> > [3] Otmakhova, Y., Verspoor, K.M., & Lau, J.H. (2022). Cross-linguistic Comparison of Linguistic Feature Encoding in BERT Models for Typologically Different Languages. Proceedings of the 4th Workshop on Research in Computational Linguistic Typology and Multilingual NLP.
> >
> > [4] Philippy, F., Guo, S., & Haddadan, S. (2023). Identifying the Correlation Between Language Distance and Cross-Lingual Transfer in a Multilingual Representation Space. ArXiv, abs/2305.02151.
> >
> > [5] Chang, T.A., Tu, Z., & Bergen, B.K. (2022). The Geometry of Multilingual Language Model Representations. Conference on Empirical Methods in Natural Language Processing.
> >
> > [6] Rafferty, A.N., & Manning, C.D. (2008). Parsing Three German Treebanks: Lexicalized and Unlexicalized Baselines.
> >
> > [7] Chang, P., Tseng, H., Jurafsky, D., & Manning, C.D. (2009). Discriminative Reordering with Chinese Grammatical Relations Features. SSST@HLT-NAACL.
> >
> > [8] Xia, M., & Monti, E. (2021). Multilingual Neural Semantic Parsing for Low-Resourced Languages. ArXiv, abs/2106.03469.
> >
> > [9] Celikyilmaz, A., Clark, E., & Gao, J. (2020). Evaluation of Text Generation: A Survey. ArXiv, abs/2006.14799.
> >
> > [10] OpenAI (2023). GPT-4 Technical Report. ArXiv, abs/2303.08774.
> >
> > [11] Raunak, V., Sharaf, A., Awadallah, H.H., & Menezes, A. (2023). Leveraging GPT-4 for Automatic Translation Post-Editing. ArXiv, abs/2305.14878.
> >
> > [12] Lee, J., Liu, A., Ahia, O., Gonen, H., & Smith, N.A. (2023). That was the last straw, we need more: Are Translation Systems Sensitive to Disambiguating Context? ArXiv, abs/2310.14610.
> >
> > [13] https://cloud.google.com/vertex-ai/docs/generative-ai/image/image-captioning
> >
> > [14] Erdem, Erkut, et al. "Neural natural language generation: A survey on multilinguality, multimodality, controllability and learning." Journal of Artificial Intelligence Research 73 (2022): 1131-1207.
> >
> > [15] Wang, Xin, et al. "Vatex: A large-scale, high-quality multilingual dataset for video-and-language research." Proceedings of the IEEE/CVF International Conference on Computer Vision. 2019
> >
> > [16] Srinivasan, Krishna, et al. "Wit: Wikipedia-based image text dataset for multimodal multilingual machine learning." Proceedings of the 44th International ACM SIGIR Conference on Research and Development in Information Retrieval. 2021.
> >
> > [17] Liu, Fangyu, et al. "Visually Grounded Reasoning across Languages and Cultures." Proceedings of the 2021 Conference on Empirical Methods in Natural Language Processing. 2021.

---

### Official Review · Reviewer_qG2w · 2023-11-01

**Soundness:** 2 fair
**Presentation:** 3 good
**Contribution:** 3 good
**Rating:** 6
**Confidence:** 4

**Summary:**

The paper delves into an in-depth analysis of semantic concept diversity within vision-language datasets, models, and their applications, spanning a variety of languages.

To achieve this, the authors have developed metrics specifically tailored for gauging semantic coverage in image descriptions across multiple languages. Utilising these metrics, they conduct a comprehensive evaluation of various existing datasets, including those generated by models and commercial API image descriptions.

The results suggest the significant impact of language and culture on image description, revealing that individuals from different linguistic and cultural backgrounds tend to highlight distinct semantic content within images. Additionally, models trained on monolingual datasets demonstrate optimal results when tested in the same language, whereas those trained on multilingual data consistently exhibit robust performance across a diverse range of test sets. This phenomenon underscores the extent to which linguistic differences in image descriptions are internalised and reproduced by the models.

**Strengths:**

+ The paper touches upon a pressing issue in vision-language research, that is the semantic content diversity of image descriptions in different languages.

+ The paper has done a good job investigating image caption’s linguistic diversity in a comprehensive way. The metrics cover different aspects of semantic content. Existing datasets, model behaviour, and commercial API outputs are all covered in the study.

**Weaknesses:**

- Certain findings presented in the paper may be contextually constrained by the scale of the experiments conducted. For instance, in the case of the fine-tuning experiment, it is plausible that the results observed may not extrapolate seamlessly to scenarios of much larger scale, such as foundation model pretraining. Given this potential limitation, a more cautious and nuanced articulation of these conclusions might be more suitable.

- Some segments of the research are founded on datasets of relatively modest size. For instance, the dataset analysis is based on 30 descriptions across two languages (English & Japanese), contributed by 10 individuals per language. This sample size may not fully encapsulate the breadth of diversity and nuance inherent to linguistic expression, potentially limiting the representativeness of the findings.

**Questions:**

* Regarding the diversity of the scene graph, I'm curious if issues in normalization could give the impression that multilingual descriptions contain more concepts. Even though you've included a step for canonicalization, can you confirm whether this process accurately aligns 100% of identical concepts? Is it possible that some concepts from multilingual sources, once translated to English, are interpreted as slightly different, leading to inflated diversity scores?

* On a related note, I tend to think that translation inherently introduces linguistic diversity, as it's rarely perfectly accurate. For instance, back translation is a popular method for data augmentation, despite the expectation that the core semantic content should remain consistent. Could you address the possibility that the observed differences in your study are simply artifacts of the translation process? I understand that human evaluation was conducted for GPT-4 translations, but subtle nuances may still have been overlooked.

---

> ### Author Response · Authors · 2023-11-21
>
> We are grateful to the reviewer for taking the time to provide valuable feedback on our work. We address below the questions raised in the review:
>
> 1. “Regarding the diversity of the scene graph, could issues in normalization give the impression that multilingual descriptions contain more concepts?”
>
> This is certainly a possibility, and we appreciate the reviewer for bringing it up. Concept alignment will always be a messy process. As such, we designed our canonicalization process to err on the side of grouping two concepts together as opposed to apart. In our canonicalization process, we first evaluate WordNet similarity (the commonly used procedure). If it fails, we evaluate similarity of embeddings. If even this fails, we look towards other associated information (e.g. attributes and relations from objects) to make a match if possible, with even lower thresholds for similarity. Therefore, we believe that our results stand despite, rather than in part because of, irreducible error in concept alignment.
>
> 2. “Could you address the possibility that the observed differences in your study are simply artifacts of the translation process?”
>
> It is true that translation introduces some minimal amount of error or diversity into text. However, we are principally concerned with the content (objects, attributes, relations) of captions, which we confirmed in our human studies are preserved across translation. Consider the example of a bird in the background. A caption which mentions a bird will result in a translation which has a bird, and a caption which does not will result in a translation which does not have a bird. Therefore, we believe the signal we are measuring (the presence of objects, attributes, relations) is not likely to be sensitive to artifacts introduced during translation.
>
> It is possible that translation artifacts may have a larger effect on the representational and linguistic diversity measures, which evaluate over raw text rather than scene graphs, which are parsed for content. However, observe from Tables 4 and 5 that the diversity metrics across all 7 languages increases substantially over the diversity metrics from groups of 3 languages. If the text was more or less the same and translation was introducing most of the diversity, we would not expect as much of an increase. Likewise, observe that the diversity metrics for groups of 3 non-English (i.e. all translated) languages are generally similar to the diversity metrics for groups of 1 English and 2 non-English languages. All of this is to suggest that although there may be an irreducible translation error, we have good reason to believe it does not significantly affect our diversity metrics.

---

> > ### Author Response · Authors · 2023-11-21
> >
> > In response to each of the weaknesses raised:
> >
> > 1. “Certain findings presented in the paper may be contextually constrained by the scale of the experiments conducted. Given this potential limitation, a more cautious and nuanced articulation of these conclusions might be more suitable.”
> >
> > We appreciate this note. We have modified our language and claims to be more nuanced throughout the paper. We are currently working in our future work on applying these results at a large scale over LAION.
> >
> > 2. “Some segments of the research are founded on datasets of relatively modest size. For instance, the dataset analysis is based on 30 descriptions across two languages (English & Japanese), contributed by 10 individuals per language. This sample size may not fully encapsulate the breadth of diversity and nuance inherent to linguistic expression, potentially limiting the representativeness of the findings.”
> >
> > Indeed, our user study recruiting American and Japanese participants to caption images was fairly limited in size. However, it should be noted that this user study was not a main result but rather an attempt to reproduce some of the behavior observed in cross-cultural psychology experiments (e.g. [1]). We base our claims off of the fundamental results of the paper, which are conducted over fairly large datasets. We analyze [3600 images × 7 languages × 3 captions each × 3 sources (original dataset, Vertex API, LLaVA) = 226,800] captions in CrossModal and [2000 images × 7 languages × 3 captions each × 3 sources (original dataset, Vertex API, LLaVA) = 126,000] captions in Visual Genome. However, we are pursuing future work to determine if this still holds in even more general contexts, such as web-scraped datasets like LAION.
> >
> > We hope this helps clarify! Please let us know if there are any other comments or questions which may arise. If we addressed your concerns, we would appreciate an increase in your score.
> >
> > – the authors
> >
> >
> >
> >
> > References
> >
> > [1] Masuda, T., & Nisbett, R. E. (2001). Attending holistically versus analytically: comparing the context sensitivity of Japanese and Americans. Journal of personality and social psychology, 81(5), 922–934. https://doi.org/10.1037//0022-3514.81.5.922

---

### Official Review · Reviewer_DPPs · 2023-11-01

**Soundness:** 3 good
**Presentation:** 3 good
**Contribution:** 3 good
**Rating:** 8
**Confidence:** 4

**Summary:**

This paper checks the hypothesis that multilingual captions of visual scenes will be more diverse than monolingual ones. This is tested both for captions produced by trained captioning models as well as with human annotators. The languages used in the study are Indo-European (en, de, fr, ru) and East Asian (zh, ja, ko). According to several metrics of caption diversity, it is indeed the case that multilingual captions are more diverse, more so for models than for humans.

**Strengths:**

The methods used in the paper are straightforward and the presentation of methods and results easy to follow. Multiple metrics are used in an exhaustive set of experiments. A reasonably diverse set of languages is tested.

**Weaknesses:**

The paper does not have any fatal or major weaknesses.

The aspect which I found disappointing was that while the introductory sections make a big deal out of cross-cultural differences, especially between European (Western) and East Asian cultures, in the empirical part of the paper there is no in-depth analysis which would shed light on specific sources of the increase of diversity due to multilingual captions, and whether these are in line or not with the types of differences claimed in the social science literature. There is the off-hand observation about Japanese captions mentioning more background objects than American ones: but much more could be done here.

Minor: "linguistic diversity" doesn't seem to have all that much to do with linguistic information: it's more about psychological attributes.

**Questions:**

There seems to be a bit of a confusion regarding the reason from scene descriptions being different in different languages. In the paper you focus on the claim that "visual perception" differs between cultures. However it could also be the case (and it's even more plausible) that *perception* itself is relatively constant, but what people *choose to mention* when describing what they perceive is different. What exactly is your claim here?

---

> ### Author Response · Authors · 2023-11-21
>
> We thank the reviewer for taking time to provide valuable feedback on our work and are pleased that they found our work interesting.
>
> The reviewer makes the insightful observation that perception might be constant while what humans choose to describe might vary, and raises the question of how we use the term “visual perception”. We follow from previous work in psychology and computer vision on object salience and attention in defining what is perceived as what the subject later describes. For instance, the important work by Matusda and Nisbett (2001) [1] notably compares American and Japanese subjects’ perception by asking them to describe what they saw after shown the scene for several images. Other examples in which human documentation is taken to indicate perception and salience include [2, 3].
>
> It is possible to measure “low-level” perception or attention with eye/gaze-tracking [4], but perception also involves importance/salience-judgements which are more important to study for the purpose of computer vision. If someone’s eyes passed over some object but they didn’t register it enough to mention it when asked to describe what was in the image, did they really perceive it? For the purposes of our paper, we assume the answer is no.
>
> To clarify, we are not making claims about what people see or their gaze patterns. Instead, we are suggesting that the objects, attributes, and relations people perceive (in the sense of registering and finding salient to mention) differ across the language they speak. Indeed, cultural and linguistic structures/norms may play a role in affecting this process, which we discuss in “Linguistic Structure and Information” in Appendix A.1.

---

> ### Author Response · Authors · 2023-11-21
>
> In response to each of the weaknesses/disappointments raised:
>
> 1. Not enough focus on cross-cultural differences
>
> We agree with the reviewer. Work in psychology demonstrating cross-cultural differences in perception (e.g., [1]) were the principal motivation for our work. Many of these studies consider language as a dimension across which to study cross-cultural differences. Ultimately, we found that it is much more difficult to track “culture” in an ML context than in a closed, experimental psychology context. One reason why is because we often don’t have information about the cultural background of someone who wrote an image caption. Therefore, the focus of our paper is to demonstrate content differences across languages in general across different metrics. Our hope is that this will motivate future work which explicitly investigates cross-cultural differences in vision-language models.
>
> 2. "Linguistic diversity" doesn't seem to have all that much to do with linguistic information: it's more about psychological attributes.
>
> This may have been a communication issue on our part. The differences which we observe arise both from culture and language. Culture affects our experiences and the importance-judgements we make in perceiving the world. Language may also affect our perception in certain ways (see [5] and discussion in Appendix A.1, “Linguistic Structure and Information”), but more substantively, language is a structure which incentivizes the expression of some information and disincentivizes others. For instance, German’s complex morphosyntactic system provides events
> with nuanced understanding of spatial relationships [6].
>
> As documented by anthropologists and linguists [7, 8], it is difficult to disentangle the effects of culture and language. As such, we use language as our independent variable but study both culture and language effects indiscriminately (hence “cultural and linguistic diversity”). This approach of studying culture and language jointly has a precedent in various social sciences fields, e.g.. in [9].
>
> Note: The LIWC metrics we used are technically psychometric indicators, but importantly for us they provide linguistic information about how sentences are being structured, what information is being included or excluded, etc. See [10, 11] which use LIWC to investigate language formation styles.
>
> We hope this helps clarify! Please let us know if there are any other comments or questions which may arise.
>
> – the authors
>
>
> References
>
> [1] Masuda, T., & Nisbett, R. E. (2001). Attending holistically versus analytically: comparing the context sensitivity of Japanese and Americans. Journal of personality and social psychology, 81(5), 922–934. https://doi.org/10.1037//0022-3514.81.5.922
>
> [2] Merrielle Spain and Pietro Perona. Some objects are more equal than others: Measuring and predicting importance. In European Conference on Computer Vision, 2008.
>
> [3] Alexander C. Berg, Tamara L. Berg, Hal Daume, Jesse Dodge, Amit Goyal, Xufeng Han, Alyssa C. Mensch, Margaret Mitchell, Aneesh Sood, Karl Stratos, and Kota Yamaguchi. Understanding and predicting importance in images. 2012 IEEE Conference on Computer Vision and Pattern Recognition, pp. 3562–3569, 2012.
>
> [4] Chua, H. F., Boland, J. E., & Nisbett, R. E. (2005). Cultural variation in eye movements during scene perception. Proceedings of the National Academy of Sciences of the United States of America, 102(35), 12629–12633. https://doi.org/10.1073/pnas.0506162102
>
> [5] Lera Boroditsky. Linguistic relativity. Encyclopedia of cognitive science, 2006.
>
> [6] Jakob Prange and Nathan Schneider. Draw mir a sheep: A supersense-based analysis of german case and adposition semantics. KI-Kunstliche Intelligenz , 35(3-4):291–306, 2021.
>
> [7] Goldstein, L. J. (1957). On Defining Culture. American Anthropologist, 59(6), 1075–1081. http://www.jstor.org/stable/666466
>
> [8] Lazear, E. P. (1999). Culture and Language. Journal of Political Economy, 107(S6), S95–S126. https://doi.org/10.1086/250105
>
> [9] Laesser, Christian & Beritelli, Pietro & Heer, Samuel. (2014). Different native languages as proxy for cultural differences in travel behaviour: Insights from multilingual Switzerland. International Journal of Culture. 8. 10.1108/IJCTHR-02-2014-0010.
>
> [10] Qiu, L., Lin, H., Ramsay, J.E., & Yang, F. (2012). You are what you tweet: Personality expression and perception on Twitter. Journal of Research in Personality, 46, 710-718.
>
> [11] Ireland, M., & Pennebaker, J.W. (2010). Language style matching in writing: synchrony in essays, correspondence, and poetry. Journal of personality and social psychology, 99 3, 549-71.

---

### Meta-Review · Area_Chair_rjsp · 2023-12-10

**Metareview:**

This paper presents an analysis on the difference in information that different languages showcase either through human raters, or models. The difference in information is measured through syntactic/semantic tags calculations in the translation of the non-English languages back in English. The concept behind the work is very interesting and it might even be true. But I think the paper needs more empirical analysis to really establish this as its quite a big claim. For example, I wonder if the differences in information in different languages, is merely a difference in that we have different humans doing these annotations. IOW what if we take more and more English native speakers and ask them to generate caption for an image? Would that also increase the information content for a picture? This is what we will need to contrast against having more annotators from a different language. Re the models, I am unsure if models in a language can significantly diverge from the raters in the language, thus addressing the information divergence in raters is more essential. (Yes, pretrained LLMs in different languages could have different tendencies but finetuning is often the major factor in the final behavior of the model).

**Justification For Why Not Higher Score:**

The empirical evidence presented in the paper is quite weak. This would be a great short paper at *ACL if the analysis is done better.

**Justification For Why Not Lower Score:**

n/a

---

### Decision · Program_Chairs · 2024-01-16

Reject